# Energy-Based Learning Algorithms
# for Analog Computing:
# A Comparative Study

**Benjamin Scellier**
Rain AI
benjamin@rain.ai

**Maxence Ernoult**
Rain AI
maxence@rain.ai

**Jack Kendall**
Rain AI
jack@rain.ai

**Suhas Kumar**
Rain AI
suhas@rain.ai

## Abstract

Energy-based learning algorithms have recently gained a surge of interest due to their compatibility with analog (post-digital) hardware. Existing algorithms include contrastive learning (CL), equilibrium propagation (EP) and coupled learning (CpL), all consisting in contrasting two states, and differing in the type of perturbation used to obtain the second state from the first one. However, these algorithms have never been explicitly compared on equal footing with same models and datasets, making it difficult to assess their scalability and decide which one to select in practice. In this work, we carry out a comparison of seven learning algorithms, namely CL and different variants of EP and CpL depending on the signs of the perturbations. Specifically, using these learning algorithms, we train deep convolutional Hopfield networks (DCHNs) on five vision tasks (MNIST, F-MNIST, SVHN, CIFAR-10 and CIFAR-100). We find that, while all algorithms yield comparable performance on MNIST, important differences in performance arise as the difficulty of the task increases. Our key findings reveal that negative perturbations are better than positive ones, and highlight the centered variant of EP (which uses two perturbations of opposite sign) as the best-performing algorithm. We also endorse these findings with theoretical arguments. Additionally, we establish new SOTA results with DCHNs on all five datasets, both in performance and speed. In particular, our DCHN simulations are 13.5 times faster with respect to Laborieux et al. [2021], which we achieve thanks to the use of a novel energy minimisation algorithm based on asynchronous updates, combined with reduced precision (16 bits).

## 1 Introduction

Prior to the dominance of backpropagation-based machine learning, Hopfield, Hinton and others proposed an alternative 'energy-based' learning (EBL) approach [Hopfield, 1984, Hinton et al., 1984]. In this approach, the learning model is described by a state variable whose dynamics is governed by an energy function. An EBL model can be used to compute as follows: 1) clamp the model input, 2) let the model settle to equilibrium (that is, the configuration of lowest energy), and 3) read the model output. The objective is to modify the weights of the model so that it computes a desired input-to-output function. This is achieved thanks to an EBL algorithm. One of the earliest EBL algorithms was constrastive learning (CL)[Movellan, 1991, Baldi and Pineda, 1991], which adjusts the model weights by contrasting two states: a 'free' state where the model outputs are free, and a perturbed (or 'clamped') state where the model outputs are clamped to desired values. In the machine learning literature, interest in EBL algorithms has remained limited due to the widespread success of backpropagation running on graphics processing units (GPUs). However, EBL algorithms have more recently revived interest as a promising learning framework for *analog* learning machines [Kendall

et al., 2020, Stern et al., 2021]. The benefit of these algorithms is that they use a single computational circuit or network for both inference and training, and they rely on local learning rules (i.e. the weight updates are local). The locality of computation and learning makes these algorithms attractive for training adaptive physical systems in general [Stern and Murugan, 2023], and for building energy-efficient analog AI in particular [Kendall et al., 2020]. Small-scale EBL-trained variable resistor networks have already been built [Dillavou et al., 2022, 2023, Yi et al., 2023], projecting a possible $10,000\times$ improvement in energy efficiency compared to GPU-based training of deep neural networks [Yi et al., 2023].

In recent years, various EBL algorithms have been proposed, such as equilibrium propagation (EP) [Scellier and Bengio, 2017], the centered variant of EP [Laborieux et al., 2021] and coupled learning (CpL) [Stern et al., 2021]. These algorithms, which are variants of CL with modified perturbation methods, are often evaluated on different models and different datasets without being compared to CL and to one another.[1] Due to the lack of explicit comparison between these algorithms, and since the algorithmic differences between them are small, they are often gathered under the 'contrastive learning' (or 'contrastive Hebbian learning') umbrella name [Dillavou et al., 2022, Stern and Murugan, 2023, Peterson and Lavin, 2022, Lillicrap et al., 2020, Luczak et al., 2022, Høier et al., 2023]. Consequently, many recent follow-up works in energy-based learning indifferently pick one of these algorithms without considering alternatives [Dillavou et al., 2022, Wycoff et al., 2022, Stern et al., 2022, Kiraz et al., 2022, Watfa et al., 2023, Yi et al., 2023, Dillavou et al., 2023, Altman et al., 2023]. The main contribution of the present work is to provide an explicit comparison of the above-mentioned EBL algorithms and highlight the important differences arising when the difficulty of the task increases. Nonetheless, comparing these algorithms comes with another challenge: simulations are typically very slow. Due to this slowness, EBL algorithms have often been used to train small networks on small datasets (by deep learning standards).[2] [3] Likewise, experimental realizations of EBL algorithms on analog hardware ("physical learning machines") have thus far been performed on small systems only.[4]

In this work, we conduct a study to compare seven EBL algorithms, including the four above-mentioned and three new ones. Depending on the sign of the perturbation, we distinguish between 'positively-perturbed' (P-), 'negatively-perturbed' (N-) and 'centered' (C-) algorithms. To avoid the problem of slow simulations of analog circuits, we conduct our comparative study on deep convolutional Hopfield networks (DCHNs)[Ernoult et al., 2019], an energy-based network that has been emulated on Ising machines [Laydevant et al., 2023], and for which simulations are faster and previously demonstrated to scale to tasks such as CIFAR-10 [Laborieux et al., 2021] and Imagenet 32x32 [Laborieux and Zenke, 2022]. Our contributions include the following:

- We train DCHNs with each of the seven EBL algorithms on five vision tasks: MNIST, Fashion-MNIST, SVHN, CIFAR-10 and CIFAR-100. We find that all these algorithms perform well on MNIST, but as the difficulty of the task increases, important behavioural differences start to emerge. Perhaps counter-intuitively, we find that N-type algorithms outperform P-type algorithms by a large margin on most tasks. The C-EP algorithm emerges as the best performing one, outperforming the other six algorithms on the three hardest tasks (SVHN, CIFAR-10 and CIFAR-100).

- We state novel theoretical results for EP, adapted from those of 'agnostic EP' [Scellier et al., 2022], that support our empirical findings. While EP is often presented as an algorithm that approximates gradient descent on the cost function [Scellier and Bengio, 2017, Laborieux et al., 2021], we provide a more precise and stronger statement: EP performs (exact) gradient descent on a surrogate function that approximates the (true) cost function. The surrogate

---

[1] For example, Kendall et al. [2020] and Watfa et al. [2023] train layered nonlinear resistive networks using EP, while Stern et al. [2021] train randomly-connected elastic and flow networks using CpL.

[2] Kendall et al. [2020] use SPICE to simulate the training of a one-hidden-layer network (with 100 'hidden nodes') on MNIST, which takes one week for only ten epochs of training. Watfa et al. [2023] train resistive networks with EP on Fashion-MNIST, Wine, and Iris. Similarly, Stern et al. [2021] simulate the training of disordered networks of up to 2048 nodes on a subset of 200 images of the MNIST dataset.

[3] With some exceptions, e.g. Laborieux and Zenke [2022] perform simulations of energy-based convolutional networks (DCHNs) on ImageNet 32x32.

[4] Dillavou et al. [2022] train a resistive network of 9 nodes and 16 edges on the IRIS dataset. Yi et al. [2023] train a 24-33-7 memristive network on a $64 \times 64$ array of memristors to classify Braille words. Laydevant et al. [2023] train a 784-120-40 network (Ising machine) on a subset of 1000 images of the MNIST dataset.

function of N-EP is an upper bound of the cost function, whereas the one of P-EP is a lower bound (Theorem 2). Moreover, the surrogate function of C-EP approximates the true cost function at the second order in the perturbation strength (Theorem 3), whereas the ones of P-EP and N-EP are first-order approximations.

- We achieve state-of-the-art DCHN simulations on all five datasets, both in terms of performance (accuracy) and speed. For example, a full run of 100 epochs on CIFAR-10 yields a test error rate of 10.4% and takes 3 hours 18 minutes on a single A100 GPU ; we show that this is 13.5x faster than the simulations of Laborieux et al. [2021] on the same hardware (a A100 GPU). With further training (300 epochs), the test error rate goes down to 9.7% – to be compared with 11.4% reported in Laborieux and Zenke [2022]. Our simulation speedup is enabled thanks to the use of a novel energy minimization procedure for DCHNs based on asynchronous updates and the use of 16 bit precision.

We note that our work also bears similarities with the line of works on energy-based models [Grathwohl et al., 2019, Nijkamp et al., 2019, Du and Mordatch, 2019, Geng et al., 2021]. Like our approach, these studies involve the minimization of an energy function within the activation space (or input space) of a network. However, unlike our approach, they do not typically exclude the use of the backpropagation algorithm, employing it not only to compute the parameter gradients, but also to execute gradient descent within the network's activation space (or input space). In contrast, the primary motivation of our work is to eliminate the need for backpropagation, and to perform inference and learning by leveraging locally computed quantities, with the long term goal of building energy-efficient processors dedicated to model optimization. Hence, our motivation diverges significantly from traditional 'energy-based models'.

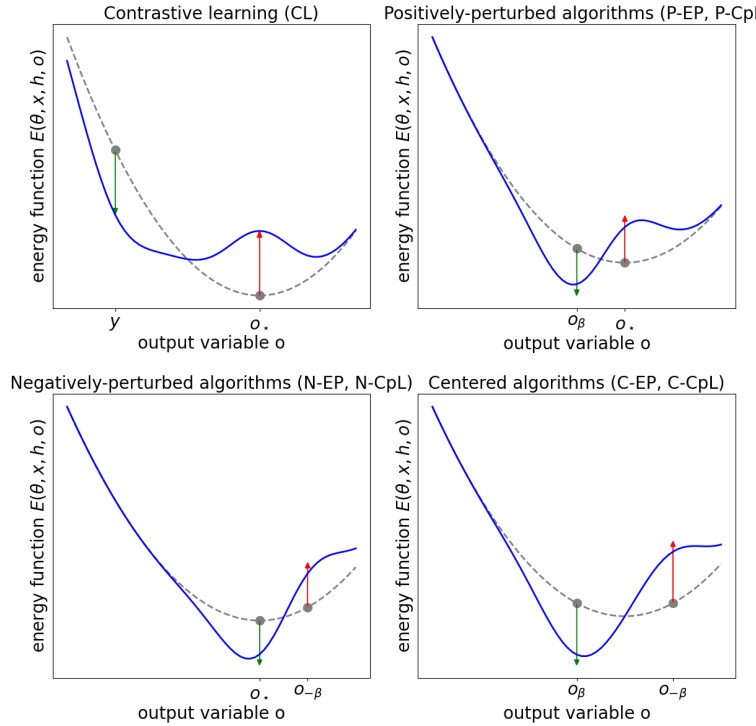

Figure 1: Cartoon illustrating the seven energy-based learning (EBL) algorithms: contrastive learning (CL), positively-perturbed algorithms (P-), negatively-perturbed algorithms (N-) and centered algorithms (C-). EP and CpL stand for equilibrium propagation and coupled learning, respectively. The desired output is $y$. The model prediction is $o_\star$ i.e. the output configuration minimizing the energy function. The strength of the perturbation is $\beta$. A positive perturbation pulls the model output ($o_\beta$) towards $y$. A negative perturbation pushes the model output ($o_{-\beta}$) away from $y$. Arrows indicate the weight update: green (resp. red) arrows decrease (resp. increase) the energy value of the corresponding configuration.

## 2 Energy-based learning algorithms

This section introduces the concepts and notations, presents the seven energy-based learning algorithms and states the theoretical results.

We consider the setting of image classification. In this setting, an energy-based learning (EBL) model is composed of an input variable ($x$), a parameter variable ($\theta$) a hidden variable ($h$) and an output variable ($o$). A scalar function $E$ called *energy function* assigns to each tuple ($\theta, x, h, o$) a real number $E(\theta, x, h, o)$. Given $\theta$ and $x$, among all possible configurations ($h, o$), the effective configuration of the model is the equilibrium state (or steady state), denoted ($h_\star, o_\star$) which is *implicitly* defined as a minimum of the energy function,

$$(h_\star, o_\star) := \underset{(h,o)}{\arg\min} \; E(\theta, x, h, o). \tag{1}$$

The equilibrium value (or steady state value) of output variables, $o_\star$, represents a prediction of the model, which we also denote $o(\theta, x) = o_\star$ to emphasize that it depends on the input $x$ and the model parameter $\theta$.

The goal of training an EBL model is to adjust $\theta$ so that, for any input $x$, the output $o(\theta, x)$ coincides with a desired output $y$ (the label associated to $x$). We refer to an algorithm for training an EBL model as an *EBL algorithm*.

**Remark.** In the terminology of Stern and Murugan [2023], $h$ and $o$ are the 'physical degrees of freedom', $\theta$ is the set of 'learning degrees of freedom' and $E$ is the 'physical cost function'.

### 2.1 Contrastive learning

Contrastive learning (CL) is the earliest EBL algorithm [Movellan, 1991]. The CL algorithm proceeds in two phases. In the first phase, input variables $x$ are clamped, while the hidden and output variables are free to stabilize to the energy minimum ($h_\star, o_\star$) as in (1). In the second phase, the output variables $o$ are now also clamped to the desired output $y$, and the hidden variables $h$ are free to stabilize to a second energy minimum, denoted $h_\star^{\mathrm{CL}}$, characterized by

$$h_\star^{\mathrm{CL}} := \underset{h}{\arg\min} \; E(\theta, x, h, y). \tag{2}$$

The contrastive learning rule for the model parameters reads

$$\Delta^{\mathrm{CL}}\theta = \eta \left( \frac{\partial E}{\partial \theta}(\theta, x, h_\star, o_\star) - \frac{\partial E}{\partial \theta}(\theta, x, h_\star^{\mathrm{CL}}, y) \right), \tag{3}$$

where $\eta$ is a learning rate. The CL rule is illustrated in Figure 1.

### 2.2 Equilibrium propagation

Equilibrium propagation (EP) is another EBL algorithm [Scellier and Bengio, 2017]. One notable difference in EP is that one explicitly introduces a cost function $C(o, y)$ that represents the discrepancy between the output $o$ and desired output $y$. In its original formulation, EP is a variant of CL which also consists of contrasting two states. In the first phase, similar to CL, input variables are clamped while hidden and output variables are free to settle to the free state ($h_\star, o_\star$) characterized by (1). In the second phase, in contrast with CL, EP proceeds by only perturbing (or *nudging*) the output variables rather than clamping them. This is achieved by augmenting the model's energy by a term $\beta C(o, y)$, where $\beta \in \mathbb{R}$ is a scalar – the *nudging parameter*. The model settles to another equilibrium state, the perturbed state, characterized by

$$(h_\beta^{\mathrm{EP}}, o_\beta^{\mathrm{EP}}) = \underset{(h,o)}{\arg\min} \; [E(\theta, x, h, o) + \beta C(o, y)]. \tag{4}$$

In its classic form, the learning rule of EP is similar to CL:

$$\Delta^{\mathrm{EP}}\theta = \frac{\eta}{\beta} \left( \frac{\partial E}{\partial \theta}(\theta, x, h_\star, o_\star) - \frac{\partial E}{\partial \theta}(\theta, x, h_\beta^{\mathrm{EP}}, o_\beta^{\mathrm{EP}}) \right). \tag{5}$$

The EP learning rule (5) comes in two variants depending on the sign of $\beta$. In Scellier and Bengio [2017], EP is introduced in the variant with $\beta > 0$, which we call here *positively-perturbed* EP (P-EP).

In this work, we introduce the variant with $\beta < 0$, which we call *negatively-perturbed* EP (N-EP). [5] We also consider the variant of EP introduced by Laborieux et al. [2021] whose learning rule reads

$$\Delta^{\mathrm{C-EP}}\theta = \frac{\eta}{2\beta}\left(\frac{\partial E}{\partial \theta}\left(\theta, x, h_{-\beta}^{\mathrm{EP}}, o_{-\beta}^{\mathrm{EP}}\right) - \frac{\partial E}{\partial \theta}\left(\theta, x, h_{\beta}^{\mathrm{EP}}, o_{\beta}^{\mathrm{EP}}\right)\right), \qquad (6)$$

which we call *centered* EP (C-EP). The EP learning rules are illustrated in Figure 1.

## 2.3 Coupled learning

Coupled learning (CpL) is another variant of contrastive learning [Stern et al., 2021]. In the first phase, similar to CL and EP, input variables are clamped, while hidden and output variables are free to settle to their free state value $(h_\star, o_\star)$ characterized by (1). In the second phase, output variables are clamped to a weighted mean of $o_\star$ and $y$, and the hidden variables are allowed to settle to their new equilibrium value. Mathematically, the new equilibrium state is characterized by the following formulas, where the weighted mean output $(o_\beta^{\mathrm{CpL}})$ is parameterized by a factor $\beta \in \mathbb{R} \setminus \{0\}$:

$$h_\beta^{\mathrm{CpL}} := \underset{h}{\arg\min}\, E(\theta, x, h, o_\beta^{\mathrm{CpL}}), \qquad o_\beta^{\mathrm{CpL}} := (1 - \beta)o_\star + \beta y. \qquad (7)$$

Similarly to CL and EP, the learning rule of CpL reads

$$\Delta^{\mathrm{CpL}}\theta = \frac{\eta}{\beta}\left(\frac{\partial E}{\partial \theta}\left(\theta, x, h_\star, o_\star\right) - \frac{\partial E}{\partial \theta}\left(\theta, x, h_\beta^{\mathrm{CpL}}, o_\beta^{\mathrm{CpL}}\right)\right). \qquad (8)$$

In particular, for $\beta = 1$, one recovers the contrastive learning algorithm (CL). In their original formulation, Stern et al. [2021] use $\beta > 0$ ; here we refer to this algorithm as *positively-perturbed* CpL (P-CpL). Similarly to EP, we also introduce *negatively-perturbed* CpL (N-CpL, with $\beta < 0$) as well as *centered* CpL (C-CpL):

$$\Delta^{\mathrm{C-CpL}}\theta = \frac{\eta}{2\beta}\left(\frac{\partial E}{\partial \theta}\left(\theta, x, h_{-\beta}^{\mathrm{CpL}}, o_{-\beta}^{\mathrm{CpL}}\right) - \frac{\partial E}{\partial \theta}\left(\theta, x, h_\beta^{\mathrm{CpL}}, o_\beta^{\mathrm{CpL}}\right)\right). \qquad (9)$$

Figure 1 also depicts P-CpL, N-CpL and C-CpL.

## 2.4 Theoretical results

The EBL algorithms presented above have different theoretical properties. We start with CL.

**Theorem 1** (Contrastive learning)**.** *The contrastive learning rule* (3) *performs one step of gradient descent on the so-called contrastive function* $\mathcal{L}_{\mathrm{CL}}$,

$$\Delta^{\mathrm{CL}}\theta = -\eta\frac{\partial \mathcal{L}_{\mathrm{CL}}}{\partial \theta}(\theta, x, y), \qquad \mathcal{L}_{\mathrm{CL}}(\theta, x, y) := E\left(\theta, x, h_\star^{\mathrm{CL}}, y\right) - E\left(\theta, x, h_\star, o_\star\right). \qquad (10)$$

Theorem 1 is proved in Movellan [1991]. However, it is not clear that the contrastive function $\mathcal{L}_{\mathrm{CL}}$ has the desirable properties of an objective function from a machine learning perspective.

The equilibrium propagation (EP) learning rules have better theoretical properties. EP is often presented as an algorithm that approximates the gradient of the cost function, e.g. in Scellier and Bengio [2017], Laborieux et al. [2021]. In this work, we provide a more precise and stronger statement: EP computes the exact gradient of a surrogate function that approximates the cost function. Moreover, in N-EP, the surrogate function is an upper bound of the cost function (Theorem 2), and in C-EP, the surrogate function approximates the cost function at the second order in $\beta$ (Theorem 3). Theorems 2 and 3 are adapted from Scellier et al. [2022] [6] – see Appendix A for proofs.

---

[5]Another variant of EP proposed in Scellier and Bengio [2017] combines N-EP and P-EP, where the sign of the nudging parameter $\beta$ is chosen at random for each example in the training set. However, N-EP wasn't used and tested as is.

[6]In 'agnostic equilibrium propagation' (AEP) [Scellier et al., 2022], the function $\mathcal{L}_\beta^{\mathrm{EP}}$ is proved to be a Lyapunov function for AEP, but AEP does not perform exact gradient descent on $\mathcal{L}_\beta^{\mathrm{EP}}$. In contrast, EP performs exact gradient descent on $\mathcal{L}_\beta^{\mathrm{EP}}$, but $\mathcal{L}_\beta^{\mathrm{EP}}$ is not a Lyapunov function for EP.

**Theorem 2** (Equilibrium propagation). *There exists some function $\mathcal{L}_\beta^{\mathrm{EP}}$ such that the learning rule* (5) *performs one step of gradient descent on $\mathcal{L}_\beta^{\mathrm{EP}}$:*

$$\Delta^{\mathrm{EP}}\theta = -\eta \frac{\partial \mathcal{L}_\beta^{\mathrm{EP}}}{\partial \theta}(\theta, x, y). \tag{11}$$

*The function $\mathcal{L}_\beta^{\mathrm{EP}}$ is a lower bound of the 'true' cost function if $\beta > 0$ (P-EP), and an upper bound if $\beta < 0$ (N-EP), i.e.*

$$\mathcal{L}_\beta^{\mathrm{EP}}(\theta, x, y) \leq C\left(o(\theta, x), y\right) \leq \mathcal{L}_{-\beta}^{\mathrm{EP}}(\theta, x, y), \qquad \forall \beta > 0, \tag{12}$$

*where $o(\theta, x) = o_\star$ is the free equilibrium value of output variables given $\theta$ and $x$, as in* (1). *Furthermore, $\mathcal{L}_\beta^{\mathrm{EP}}$ approximates the 'true' cost function up to $O(\beta)$ when $\beta \to 0$,*

$$\mathcal{L}_\beta^{\mathrm{EP}}(\theta, x, y) = C\left(o(\theta, x), y\right) + O(\beta). \tag{13}$$

**Theorem 3** (Centered EP). *There exists some function $\mathcal{L}_{-\beta;+\beta}^{\mathrm{EP}}$ such that the learning rule* (6) *performs one step of gradient descent on $\mathcal{L}_{-\beta;+\beta}^{\mathrm{EP}}$:*

$$\Delta^{\mathrm{C-EP}}\theta = -\eta \frac{\partial \mathcal{L}_{-\beta;+\beta}^{\mathrm{EP}}}{\partial \theta}(\theta, x, y). \tag{14}$$

*Moreover, the function $\mathcal{L}_{-\beta;+\beta}^{\mathrm{EP}}$ approximates the 'true' cost function up to $O(\beta^2)$ when $\beta \to 0$,*

$$\mathcal{L}_{-\beta;+\beta}^{\mathrm{EP}}(\theta, x, y) = C\left(o(\theta, x), y\right) + O(\beta^2). \tag{15}$$

Finally, the analysis of the coupled learning rules (P-CpL, N-CpL and C-CpL) is more complicated. Stern et al. [2021] introduce the following function as a candidate loss function for coupled learning:

$$\mathcal{L}_{\mathrm{CpL}}^{(1)}(\theta, x, y) := (y - o(\theta, x))^\top \cdot \frac{\partial^2 G}{\partial o^2}(\theta, x, o(\theta, x)) \cdot (y - o(\theta, x)), \quad G(\theta, x, o) := \min_h E(\theta, x, h, o). \tag{16}$$

Stern et al. [2021] argue that the P-CpL rule (8) optimizes both $\mathcal{L}_{\mathrm{CpL}}$ and the squared error $\mathcal{L}_{\mathrm{MSE}}(\theta, x, y) := (o(\theta, x) - y)^2$. Besides, Stern et al. [2022, 2023] argue that the learning rule (8) performs gradient descent on

$$\mathcal{L}_{\mathrm{CpL}}^{(2)} := \frac{1}{\beta}\left(E(\theta, x, h_\beta^{\mathrm{CpL}}, o_\beta^{\mathrm{CpL}}) - E(\theta, x, h_\star, o_\star)\right). \tag{17}$$

In Appendix B, we demonstrate that the coupled learning rules (8) and (9) do not optimize $\mathcal{L}_{\mathrm{CpL}}^{(1)}$ or $\mathcal{L}_{\mathrm{MSE}}$, and do not perform gradient descent on $\mathcal{L}_{\mathrm{CpL}}^{(2)}$.

## 3 Deep convolutional Hopfield networks

To compare the EBL algorithms presented in section 2, we consider the EBL model of Ernoult et al. [2019], which we call *deep convolutional Hopfield network* (DCHN). We consider specifically the network architecture of Laborieux et al. [2021].

**Network architecture.** The network has an input layer, four hidden layers, and an output layer. Since we consider classification tasks, the output layer has $M$ units, where $M$ is the number of categories for the task. Successive layers are interconnected by convolutional interactions with kernel size 3×3, padding 1, and max pooling. Except for the last hidden layer and the output layer, which are interconnected by a dense interaction.

**Energy function.** We denote the state of the network $s = (s_0, s_1, s_2, s_3, s_4, s_5)$, where $x = s_0$ is the input layer, $h = (s_1, s_2, s_3, s_4)$ is the hidden variable and $o = s_5$ is the output variable. The energy function of the network is

$$E(\theta, s) := \sum_{k=1}^{5} \frac{1}{2}\|s_k\|^2 + \sum_{k=1}^{4} E_k^{\mathrm{conv}}(w_k, s_{k-1}, s_k) + E_5^{\mathrm{dense}}(w_5, s_4, s_5) + \sum_{k=1}^{5} E_k^{\mathrm{bias}}(b_k, s_k), \tag{18}$$

where $\theta = \{w_k, b_k \mid 1 \leq k \leq 5\}$ is the set of model parameters, and $E_k^{\mathrm{conv}}$, $E_k^{\mathrm{dense}}$ and $E_k^{\mathrm{bias}}$ are energy terms defined as

$$E_k^{\mathrm{conv}} := -s_k \bullet \mathcal{P}\left(w_k \star s_{k-1}\right), \qquad E_k^{\mathrm{dense}} := -s_k^\top w_k s_{k-1}, \qquad E_k^{\mathrm{bias}} := -b_k^\top s_k. \qquad (19)$$

Specifically, $E_k^{\mathrm{conv}}$ is the energy function of a convolutional interaction between layers $k-1$ and $k$, parameterized by the kernel $w_k$ (the weights), where $\star$ is the convolution operation, $\mathcal{P}$ is the max pooling operation, and $\bullet$ is the scalar product for pairs of tensors. $E_k^{\mathrm{dense}}$ is the energy of a dense interaction between layers $k-1$ and $k$, parameterized by the $\dim(s_{k-1}) \times \dim(s_k)$ matrix $w_k$. Finally, $E_k^{\mathrm{bias}}$ is the energy term of $b_k$, the bias of layer $k$.

**Cost function.** For the equilibrium propagation learning rules, we use the squared error cost function $C(o, y) = \|o - y\|^2$, where $o = s_5$ is the output layer and $y$ is the one-hot code of the label (associated to image $x$).

**State space.** Given an input $x$, the steady state of the model is $(h_\star, o_\star) = \underset{(h,o) \in \mathcal{S}}{\arg \min} \; E(\theta, x, h, o)$, where $\mathcal{S}$ is the state space, i.e. the space of configurations for the hidden and output variables. We use $\mathcal{S} = \prod_{k=1}^4 [0,1]^{\dim(s_k)} \times \mathbb{R}^{\dim(s_5)}$, where $[0,1]$ is the closed interval of $\mathbb{R}$ with bounds 0 and 1, and $\dim(s_k)$ is the number of units in layer $k$.

**Energy minimization via asynchronous updates.** In order to compute the steady states (free state and perturbed states) of the learning algorithms presented in section 2, we use a novel procedure for DCHNs using 'asynchronous updates' for the state variables, as opposed to the 'synchronous updates' used in other works [Ernoult et al., 2019, Laborieux et al., 2021, Laydevant et al., 2021, Laborieux and Zenke, 2022]. This asynchronous procedure proceeds as follows: at every iteration, we first update the layers of even indices in parallel (the first half of the layers) and then we update the layers of odd indices (the other half of the layers). Updating all the layers once (first the even layers, then the odd layers) constitutes one 'iteration'. We repeat as many iterations as necessary until convergence to the steady state. See Appendix C for a detailed explanation of the asynchronous procedure, and for a comparison with the synchronous procedure used in other works. Although we do not have a proof of convergence of this asynchronous procedure for DCHNs when $\mathcal{P}$ is the max pooling operation, we find experimentally that it converges faster than the synchronous procedure (for which there is no proof of convergence anyway).

## 4  Simulations

### 4.1  Comparative study of EBL algorithms

**Setup.** We compare with simulations the seven energy-based learning (EBL) algorithms of section 2 on the deep convolutional Hopfield network (DCHN) of section 3. To do this, we train a DCHN on MNIST, Fashion-MNIST, SVHN, CIFAR-10 and CIFAR-100 using each of the seven EBL algorithms. We also compare the performance of these algorithms to two baselines: recurrent backpropagation (RBP) and truncated backpropagation (TBP), detailed below. For each simulation, the DCHN is trained for 100 epochs. Each run is performed on a single A100 GPU. A run on MNIST and FashionMNIST takes 3 hours 30 minutes ; a run on SVHN takes 4 hours 45 minutes ; and a run on CIFAR-10 and CIFAR-100 takes 3 hours. All these simulations are performed with the same network using the same initialization scheme and the same hyperparameters. Details of the training experiments are provided in Appendix D and the results are reported in Table 1. [7]

**First baseline: recurrent backpropagation (RBP).** As the equilibrium state $o_\star$ of an EBL model is defined *implicitly* as the minimum of an energy function, the gradient of the cost $C(o_\star, y)$ can be computed via implicit differentiation, or more specifically via an algorithm called *recurrent backpropagation* [Almeida, 1987, Pineda, 1987].

**Second baseline: truncated backpropagation (TBP).** Our second baseline uses automatic differentiation (i.e. backpropagation) as follows. First, we compute the free state $(h_\star, o_\star)$. Then, starting

---

[7]The code is available at `https://github.com/rain-neuromorphics/energy-based-learning`

from the free state $(h_0, o_0) = (h_\star, o_\star)$ we perform $K$ iterations of the fixed point dynamics ; we arrive at $(h_K, o_K)$, which is also equal to the free state. We then compute the gradient of $C(o_K, y)$ with respect to the parameters ($\theta$), without backpropagating through $(h_0, o_0)$. This is a truncated version of backpropagation, where we only backpropagate through the last $K$ iterations but not through the minimization of the energy function. This baseline is also used by Ernoult et al. [2019].

Table 1: Results obtained by training the deep convolutional Hopfield network (DCHN) of Section 3 with the EBL algorithms of Section 2: contrastive learning (CL), equilibrium propagation (EP) and coupled learning (CpL). EP and CpL are tested in their positively-perturbed (P-), negatively-perturbed (N-) and centered (C-) variants. We also report two baselines: truncated backpropagation (TBP) and recurrent backpropagation (RBP). Train and Test refer to the training and test error rates, in %. For each of these 45 experiments, we perform three runs and report the mean values. See Appendix D for the complete results with std values. The hyperparameters used for this study are reported in Table 5.

|      | MNIST | | FashionMNIST | | SVHN | | CIFAR-10 | | CIFAR-100 | |
|------|------|------|------|------|------|------|------|------|------|------|
|      | Test | Train | Test | Train | Test | Train | Test | Train | Test | Train |
| TBP  | 0.42 | 0.23 | 6.12 | 3.09 | 3.76 | 2.37 | 10.1 | 3.1 | 33.4 | 17.2 |
| RBP  | 0.44 | 0.33 | 6.28 | 3.70 | 3.87 | 3.43 | 10.7 | 5.2 | 34.4 | 18.2 |
| CL   | 0.61 | 2.39 | 10.10 | 15.49 | 6.1 | 15.8 | 31.4 | 45.2 | 71.4 | 88.6 |
| P-EP | 1.66 | 2.29 | 90.00 | 89.99 | 83.9 | 81.9 | 72.6 | 79.5 | 89.4 | 95.5 |
| N-EP | **0.42** | 0.19 | **6.22** | 3.87 | 80.4 | 81.1 | 11.9 | 6.2 | 44.7 | 40.1 |
| C-EP | 0.44 | 0.20 | 6.47 | 4.01 | **3.51** | 3.01 | **11.1** | 5.6 | **37.0** | 26.0 |
| P-CpL | 0.66 | 0.59 | 64.70 | 65.31 | 40.1 | 50.8 | 46.9 | 57.7 | 77.9 | 91.0 |
| N-CpL | 0.50 | 0.88 | 6.86 | 6.27 | 80.4 | 81.1 | 13.5 | 10.2 | 51.9 | 50.6 |
| C-CpL | 0.44 | 0.38 | 6.91 | 5.29 | 4.23 | 5.05 | 14.9 | 14.6 | 46.5 | 37.9 |

We draw several lessons from Table 1.

**Algorithms perform alike on MNIST.**    Little difference is observed in the test performance of the algorithms on MNIST, ranging from 0.42% to 0.66% test error rate for six of the seven EBL algorithms. This result confirms the current trend in the literature, often skewed towards simple tasks, which sometimes goes to treat all EBL algorithms as one and the same.

**Weak positive perturbations perform worse than strong ones.**    P-EP fails on most datasets (Fashion-MNIST, SVHN, CIFAR-10 and CIFAR-100). P-CpL (the other weakly positively-perturbed algorithm) performs better than P-EP on all tasks, but also fails on CIFAR-100 (91% training error). It is noteworthy that CL, which employs full clamping to the desired output (i.e. a strong positive perturbation), performs better than these two algorithms on *all tasks*, sometimes by a very large margin (Fashion-MNIST and SVHN).

**Negative perturbations yield better results than positive ones.**    On Fashion-MNIST, CIFAR-10 and CIFAR-100, algorithms employing a negative perturbation (N-EP and N-CpL) perform significantly better than those employing a positive perturbation (CL, P-EP and P-CpL). Theorem 2 sheds light on why N-EP performs better than P-EP: N-EP optimizes an upper bound of the cost function, whereas P-EP optimizes a lower bound. We note however that on SVHN, the results obtained with N-EP and N-CpL are much worse than CL – in Appendix E we perform additional simulations where we change the weight initialization of the network ; we find that N-EP and N-CpL generally perform much better than CL, P-EP and P-CpL, supporting our conclusion.

**Two-sided perturbations (i.e. centered algorithms) yield better results than one-sided perturbations.**    While little difference in performance between the centered (C-EP and C-CpL) and negatively-perturbed (N-EP and N-CpL) algorithms is observed on MNIST, FashionMNIST and CIFAR-10, the centered algorithms unlock training on SVHN and significantly improve the test error rate on CIFAR-100 (by $\geq 5.4\%$). Theorem 3 sheds light on why C-EP performs better than N-EP: the loss function $\mathcal{L}^{\text{EP}}_{-\beta;+\beta}$ of C-EP better approximates the cost function (up to $O(\beta^2)$) than the loss function $\mathcal{L}^{\text{EP}}_{-\beta}$ of N-EP (up to $O(\beta)$).

**The EP perturbation method yields better results than the one of CpL.** While little difference in performance is observed between C-EP and C-CpL on MNIST and FashionMNIST, C-EP significantly outperforms C-CpL on CIFAR-10 and CIFAR-100, both in terms of training error rate ($\geq 11.9\%$ difference) and test error rate ($\geq 3.8\%$ difference). The same observations hold between N-EP and N-CpL. We also note that the CpL learning rules seem to be less theoretically grounded than the EP learning rules, as detailed in Appendix B.

### 4.2 State-of-the-art DCHN simulations (performance and speed)

The comparative study conducted in the previous subsection highlights C-EP as the best EBL algorithm among the seven algorithms considered in this work. Using C-EP, we then perform additional simulations on MNIST, CIFAR-10 and CIFAR-100, where we optimize the hyperparameters of training (weight initialization, initial learning rates, number of iterations, value of the nudging parameter and weight decay) to yield the best performance. We use the hyperparameters reported in Table 5 (Appendix D). We report in Table 2 our fastest simulations (100 epochs) as well as the ones that yield the lowest test error rate (300 epochs), averaged over three runs each. We also compare our results with existing works on deep convolutional Hopfield networks (DCHNs).

Table 2: We achieve state-of-the-art results (both in terms of speed and accuracy) with C-EP-trained DCHNs on MNIST, CIFAR-10 and CIFAR-100. The results are averaged over 3 runs, and compared with the existing literature on DCHNs. Top1 (resp. Top5) refers to the test error rate for the Top1 (resp. Top5) classification task. Wall-clock time (WCT) is reported as HH:MM. The hyperparameters used for these simulations are reported in Table 5 (Appendix D).

|  | MNIST | | CIFAR-10 | | CIFAR-100 | | |
| --- | --- | --- | --- | --- | --- | --- | --- |
|  | Top1 | WCT | Top1 | WCT | Top1 | Top5 | WCT |
| Ernoult et al. [2019] | 1.02 | 8:58 | | | | | |
| Laborieux et al. [2021] | | | 11.68 | N.A. | | | |
| Laydevant et al. [2021] | 0.85 | N.A. | 13.78 | $\sim$ 120:00 | | | |
| Luczak et al. [2022] | | | 20.03 | N.A. | | | |
| Laborieux and Zenke [2022] | | | 11.4 | $\sim$ 24:00 | 38.4 | **14.0** | $\sim$ 24:00 |
| This work (100 epochs) | **0.44** | **3:30** | 10.40 | **3:18** | 34.2 | 14.2 | **3:02** |
| This work (300 epochs) | | | **9.70** | 9:54 | **31.6** | 14.1 | 9:04 |

Table 2 shows that we achieve better simulation results than existing works on DCHNs on all three datasets, both in terms of performance and speed. For instance, our 100-epoch simulations on CIFAR-10 take 3 hours 18 minutes, which is 7 times faster than those reported in [Laborieux and Zenke, 2022] (1 day), and 36 times faster than those reported in Laydevant et al. [2021] (5 days), and our 300-epoch simulations on CIFAR-10 yield 9.70% test error rate, which is significantly lower than [Laborieux and Zenke, 2022] (11.4%). On CIFAR-100, it is interesting to note that our 300-epoch simulations yield a significantly better Top-1 error rate than Laborieux and Zenke [2022] (31.6% vs 38.4%), but a slightly worse Top-5 error rate (14.1% vs 14.0%) ; we speculate that this might be due to our choice of using the mean-squared error instead of the cross-entropy loss used in Laborieux and Zenke [2022]. Since no earlier work on DCHNs has performed simulations on Fashion-MNIST and SVHN, our results reported in Table 1 are state-of-the-art on these datasets as well.

Our important speedup comes from our novel energy-minimization procedure based on "asynchronous updates", combined with 60 iterations at inference (free phase) and the use of 16-bit precision. In comparison, Laborieux et al. [2021] used "synchronous updates" with 250 iterations and 32-bit precision. We show in Appendix C that these changes result in a 13.5x speedup on the same device (a A100 GPU) without degrading the performance (test error rate).

We also stress that there still exists an important gap between our SOTA DCHN results and SOTA computer vision models. For example, Dosovitskiy et al. [2020] report 0.5% test error rate on CIFAR-10, and 5.45% top-1 test error rate on CIFAR-100.

# 5 Conclusion

Our comparative study of energy-based learning (EBL) algorithms delivers a few key take-aways: 1) simple tasks such as MNIST may suggest that all EBL algorithms work equally well, but more difficult tasks such as CIFAR-100 magnify small algorithmic differences, 2) negative perturbations yield better results than positive ones, 3) two-sided (centered) perturbations perform better than one-sided perturbations, and 4) the perturbation technique of equilibrium propagation yields better results than the one of coupled learning. These findings highlight the centered variant of equilibrium propagation (C-EP) as the best EBL algorithm among those considered in the present work, outperforming the second-best algorithm on CIFAR-100 (N-EP) by a significant margin. Our results also challenge some common views. First, it is noteworthy that CL (which uses clamping to the desired output, i.e. a strong positive perturbation) yields better results than P-CpL and P-EP (which use weak positive perturbations): this is contrary to the prescription of Stern et al. [2021] to use small positive perturbations, and opens up questions in the strong perturbation regime [Meulemans et al., 2022]. Second, it is interesting to note that CL [Movellan, 1991, Baldi and Pineda, 1991], EP [Scellier and Bengio, 2017] and CpL [Stern et al., 2021] were originally introduced around the same intuition of bringing the model output values closer to the desired outputs ; the fact that the four best-performing algorithms of our study (C-EP, N-EP, C-CpL and N-CpL) all require a negative perturbation suggests on the contrary that the crucial part in EBL algorithms is not to *pull* the model outputs towards the desired outputs, but to *push away* from the desired outputs. Third, Laborieux et al. [2021] explained the very poor performance of P-EP by it approximating the gradient of the cost function up to $O(\beta)$, but our observation that N-EP considerably outperforms P-EP while also possessing a bias of order $O(\beta)$ shows that this explanation is incomplete ; Theorem 2 provides a complementary explanation: P-EP optimizes a lower bound of the cost function, whereas N-EP optimizes an upper bound.

Our work also establishes new state-of-the-art results for deep convolutional Hopfield networks (DCHNs) on all five datasets, both in terms of performance (accuracy) and speed. In particular, thanks to the use of a novel "asynchronous" energy-minimization procedure for DCHNs, we manage to reduce the number of iterations required to converge to equilibrium to 60 - compared to 250 iterations used in Laborieux et al. [2021]. Combined with the use of 16-bit precision (instead of 32-bit), this leads our simulations to be 13.5 times faster than those of Laborieux et al. [2021] when run on the same hardware (a A100 GPU).

While our theoretical and simulation results seem conclusive, we also stress some of the limiting aspects of our study. First, our comparative study of EBL algorithms was conducted only on the DCHN model ; further studies will be required to confirm whether or not the conclusions that we have drawn extend to a broader class of models such as resistive networks [Kendall et al., 2020] and elastic networks [Stern et al., 2021]. Second, in our simulations of DCHNs, the performance of the different EBL methods depends on various hyperparameters such as the perturbation strength ($\beta$), the number of iterations performed to converge to steady state, the weight initialization scheme, the learning rates, the weight decay, etc. Since a different choice of these hyperparameters can lead to different performance, it is not excluded that a deeper empirical study could lead to slightly different conclusions.

Ultimately, while our work is concerned with *simulations*, EBL algorithms are most promising for training actual analog machines ; we believe that our findings can inform the design of such hardware as well [Dillavou et al., 2022, 2023, Yi et al., 2023, Altman et al., 2023]. Finally, we contend that our theoretical insights and experimental findings may also help better understand or improve novel EBL algorithms [Anisetti et al., 2022, Laborieux and Zenke, 2022, Hexner, 2023, Anisetti et al., 2023], and more generally guide the design of better learning algorithms for bi-level optimization problems [Zucchet and Sacramento, 2022], including the training of Lagrangian systems [Kendall, 2021, Scellier, 2021], meta-learning [Zucchet et al., 2022, Park and Simeone, 2022], direct loss minimization [Song et al., 2016] and predictive coding [Whittington and Bogacz, 2019, Millidge et al., 2022].

## Acknowledgments and Disclosure of Funding

The authors thank Mohammed Fouda, Ludmila Levkova, Nicolas Zucchet, Axel Laborieux, Nachi Stern, Sam Dillavou and Arvind Murugan for discussions. This work was funded by Rain AI. Rain AI has an interest in commercializing technologies based on brain-inspired learning algorithms.

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

# A  Loss functions of the equilibrium propagation learning rules

In this appendix, we prove Theorems 2 and 3. To do so, we state and prove Theorems 4 and 5 below, of which Theorems 2 and 3 are direct consequences.

In this appendix, the input $x$ and label $y$ are fixed, so we omit them in the notation for simplicity. For fixed $\theta$, we define for any value of $\beta$ the minimum of the augmented energy when $h$ and $o$ are free:

$$F(\beta, \theta) := \inf_{h,o} \{E(\theta, h, o) + \beta C(o)\}. \tag{20}$$

We further define the contrastive functions (or 'surrogate functions')

$$\mathcal{L}_{\beta}^{\mathrm{EP}}(\theta) := \frac{F(\beta, \theta) - F(0, \theta)}{\beta}, \qquad \beta \in \mathbb{R} \setminus \{0\}, \tag{21}$$

and

$$\mathcal{L}_{-\beta;+\beta}^{\mathrm{EP}}(\theta) := \frac{F(\beta, \theta) - F(-\beta, \theta)}{2\beta}, \qquad \beta > 0. \tag{22}$$

We also recall the definition of the steady state values of output variables:

$$o(\theta) := \arg\min_{o} \min_{h} E(\theta, h, o). \tag{23}$$

Finally, we recall that the equilibrium propagation (EP) perturbed state (4) is defined as

$$(h_{\beta}^{\mathrm{EP}}, o_{\beta}^{\mathrm{EP}}) := \arg\min_{(h,o)} \{E(\theta, h, o) + \beta C(o)\} \tag{24}$$

and that the EP learning rules (5) and (6) read

$$\Delta^{\mathrm{EP}}\theta := \frac{\eta}{\beta} \left( \frac{\partial E}{\partial \theta} (\theta, h_0^{\mathrm{EP}}, o_0^{\mathrm{EP}}) - \frac{\partial E}{\partial \theta} (\theta, h_{\beta}^{\mathrm{EP}}, o_{\beta}^{\mathrm{EP}}) \right) \tag{25}$$

and

$$\Delta^{\mathrm{C-EP}}\theta := \frac{\eta}{2\beta} \left( \frac{\partial E}{\partial \theta} (\theta, h_{-\beta}^{\mathrm{EP}}, o_{-\beta}^{\mathrm{EP}}) - \frac{\partial E}{\partial \theta} (\theta, h_{\beta}^{\mathrm{EP}}, o_{\beta}^{\mathrm{EP}}) \right). \tag{26}$$

In particular, $o_0^{\mathrm{EP}} = o(\theta)$ by definition.

**Theorem 4.** *The derivative of the function $\beta \to F(\beta, \theta)$ at $\beta = 0$ is equal to $C(o(\theta))$. In particular, as $\beta \to 0$, we have*

$$\mathcal{L}_{\beta}^{\mathrm{EP}}(\theta) = C(o(\theta)) + O(\beta) \qquad and \qquad \mathcal{L}_{-\beta;+\beta}^{\mathrm{EP}}(\theta) = C(o(\theta)) + O(\beta^2). \tag{27}$$

*Furthermore, we have for any $\beta > 0$*

$$\mathcal{L}_{\beta}^{\mathrm{EP}}(\theta) \leq C(o(\theta)) \leq \mathcal{L}_{-\beta}^{\mathrm{EP}}(\theta). \tag{28}$$

**Theorem 5.** *The equilibrium propagation learning rules satisfy*

$$\Delta^{\mathrm{EP}}\theta = -\eta \frac{\partial \mathcal{L}_{\beta}^{\mathrm{EP}}}{\partial \theta}(\theta), \qquad \Delta^{\mathrm{C-EP}}\theta = -\eta \frac{\partial \mathcal{L}_{-\beta;+\beta}^{\mathrm{EP}}}{\partial \theta}(\theta). \tag{29}$$

Theorem 4 is borrowed from Scellier et al. [2022], where these statements are proved in the context of 'agnostic equilibrium propagation'. Theorem 5 is a new contribution. For clarity to ensure the manuscript is self-contained, we provide here proofs for both Theorem 4 and Theorem 5.

First, we introduce a new notation: we introduce the function

$$\overline{F}(\beta, \theta, h, o) := E(\theta, h, o) + \beta C(o). \tag{30}$$

Using this notation, for every $\beta \in \mathbb{R}$, the first order conditions for the steady state $(h_{\beta}^{\mathrm{EP}}, o_{\beta}^{\mathrm{EP}})$ read:

$$\frac{\partial \overline{F}}{\partial h}(\beta, \theta, h_{\beta}^{\mathrm{EP}}, o_{\beta}^{\mathrm{EP}}) = 0, \qquad \frac{\partial \overline{F}}{\partial o}(\beta, \theta, h_{\beta}^{\mathrm{EP}}, o_{\beta}^{\mathrm{EP}}) = 0, \tag{31}$$

and the function $F$ (20) rewrites

$$F(\beta, \theta) = \overline{F}(\beta, \theta, h_{\beta}^{\mathrm{EP}}, o_{\beta}^{\mathrm{EP}}). \tag{32}$$

*Proof of Theorem 4.* Using the chain rule of differentiation, we have

$$\frac{\partial F}{\partial \beta}(\beta, \theta) = \frac{\partial \overline{F}}{\partial \beta}(\beta, \theta, h_\beta^{\mathrm{EP}}, o_\beta^{\mathrm{EP}}) + \underbrace{\frac{\partial \overline{F}}{\partial h}(\beta, \theta, h_\beta^{\mathrm{EP}}, o_\beta^{\mathrm{EP}}) \cdot \frac{\partial h_\beta^{\mathrm{EP}}}{\partial \beta}}_{=0} + \underbrace{\frac{\partial \overline{F}}{\partial o}(\beta, \theta, h_\beta^{\mathrm{EP}}, o_\beta^{\mathrm{EP}}) \cdot \frac{\partial o_\beta^{\mathrm{EP}}}{\partial \beta}}_{=0}.$$

(33)

Therefore

$$\frac{\partial F}{\partial \beta}(\beta, \theta) = \frac{\partial \overline{F}}{\partial \beta}(\beta, \theta, h_\beta^{\mathrm{EP}}, o_\beta^{\mathrm{EP}}) = C\left(o_\beta^{\mathrm{EP}}\right).$$

(34)

Evaluating this expression at $\beta = 0$, and using $o_0^{\mathrm{EP}} = o(\theta)$ by definition, we get

$$\frac{\partial F}{\partial \beta}(0, \theta) = C\left(o(\theta)\right),$$

(35)

which is the first statement of Theorem 4. Using a Taylor expansion of $F(\beta, \theta)$ around $\beta = 0$, we have

$$F(\beta, \theta) = F(0, \theta) + \beta C\left(o(\theta)\right) + O(\beta^2).$$

(36)

Subtracting $F(\theta, 0)$ on both sides and dividing by $\beta$, we get

$$\mathcal{L}_\beta^{\mathrm{EP}}(\theta) = \frac{F(\beta, \theta) - F(0, \theta)}{\beta} = C\left(o(\theta)\right) + O(\beta),$$

(37)

which is the first part of equation (27). Similarly, we write Taylor expansions for $F(\beta, \theta)$ and $F(-\beta, \theta)$,

$$F(\beta, \theta) = F(0, \theta) + \beta C\left(o(\theta)\right) + \frac{1}{2}\beta^2 \frac{\partial^2 F}{\partial \beta^2}(0, \theta) + O(\beta^3),$$

(38)

$$F(-\beta, \theta) = F(0, \theta) - \beta C\left(o(\theta)\right) + \frac{1}{2}\beta^2 \frac{\partial^2 F}{\partial \beta^2}(0, \theta) + O(\beta^3).$$

(39)

Subtracting the second equation from the first one, and dividing by $2\beta$, we get

$$\mathcal{L}_{-\beta;+\beta}^{\mathrm{EP}}(\theta) = \frac{F(\beta, \theta) - F(-\beta, \theta)}{2\beta} = C\left(o(\theta)\right) + O(\beta^2),$$

(40)

which is the second part of equation (27).

Next, we prove the upper bound and lower bound properties. To this end, we write for any $\beta \in \mathbb{R}$,

$$F(\beta, \theta) = \inf_{h,o}\left\{E(\theta, h, o) + \beta C(o)\right\}$$

(41)

$$\leq \inf_h\left\{E(\theta, h, o(\theta)) + \beta C\left(o(\theta)\right)\right\} = F(0, \theta) + \beta C\left(o(\theta)\right),$$

(42)

where $o(\theta)$ is the free state value of output variables. Subtracting $F(\theta, 0)$ on both sides we get

$$F(\beta, \theta) - F(\theta, 0) \leq \beta C(o(\theta)).$$

(43)

Next we divide by $\beta$. For $\beta > 0$, we get

$$\mathcal{L}_\beta^{\mathrm{EP}}(\theta) = \frac{F(\beta, \theta) - F(\theta, 0)}{\beta} \leq C(o(\theta))$$

(44)

and for $\beta < 0$ we get

$$\mathcal{L}_\beta^{\mathrm{EP}}(\theta) = \frac{F(\beta, \theta) - F(\theta, 0)}{\beta} \geq C(o(\theta)).$$

(45)

Hence the result:

$$\mathcal{L}_\beta^{\mathrm{EP}}(\theta) \leq C(o(\theta)) \leq \mathcal{L}_{-\beta}^{\mathrm{EP}}(\theta), \qquad \forall \beta > 0.$$

(46)

$\square$

Similarly, it can be shown more generally that the function $\beta \to \mathcal{L}_\beta^{\mathrm{EP}}(\theta)$ is non-increasing, or equivalently that the function $\beta \mapsto F(\beta, \theta)$ is concave.

*Proof of Theorem 5.* First, using (21), we have

$$\frac{\partial \mathcal{L}_\beta^{\text{EP}}}{\partial \theta} = \frac{1}{\beta} \left( \frac{\partial F}{\partial \theta}(\beta, \theta) - \frac{\partial F}{\partial \theta}(0, \theta) \right). \tag{47}$$

Similar to the proof of Theorem 4, using the chain rule of differentiation, we have for every $\beta \in \mathbb{R}$

$$\frac{\partial F}{\partial \theta}(\beta, \theta) = \frac{\partial \overline{F}}{\partial \theta}(\beta, \theta, h_\beta^{\text{EP}}, o_\beta^{\text{EP}}) + \underbrace{\frac{\partial \overline{F}}{\partial h}(\beta, \theta, h_\beta^{\text{EP}}, o_\beta^{\text{EP}})}_{=0} \cdot \frac{\partial h_\beta^{\text{EP}}}{\partial \theta} + \underbrace{\frac{\partial \overline{F}}{\partial o}(\beta, \theta, h_\beta^{\text{EP}}, o_\beta^{\text{EP}})}_{=0} \cdot \frac{\partial o_\beta^{\text{EP}}}{\partial \theta}.$$

$$\tag{48}$$

Here again, we have used the first-order steady state conditions for $h_\beta^{\text{EP}}$ and $o_\beta^{\text{EP}}$. Therefore

$$\frac{\partial F}{\partial \theta}(\beta, \theta) = \frac{\partial \overline{F}}{\partial \theta}(\beta, \theta, h_\beta^{\text{EP}}, o_\beta^{\text{EP}}) = \frac{\partial E}{\partial \theta}(\theta, h_\beta^{\text{EP}}, o_\beta^{\text{EP}}). \tag{49}$$

Injecting (49) in (47), we get

$$\frac{\partial \mathcal{L}_\beta^{\text{EP}}}{\partial \theta}(\theta) = \frac{1}{\beta} \left( \frac{\partial E}{\partial \theta}(\theta, h_\beta^{\text{EP}}, o_\beta^{\text{EP}}) - \frac{\partial E}{\partial \theta}(\theta, h_0^{\text{EP}}, o_0^{\text{EP}}) \right). \tag{50}$$

We conclude using the definitions of the equilibrium propagation learning rule (5). We prove similarly the case of centered EP with $\frac{\partial \mathcal{L}_{-\beta; +\beta}^{\text{EP}}}{\partial \theta}(\theta)$. □

# B    On the coupled learning rules

In this appendix, we prove some theoretical results related to the coupled learning rules.

Recall the definition of the free state

$$(h_\star, o_\star) := \arg\min_{(h,o)} E(\theta, x, h, o), \tag{51}$$

the definition of the perturbed state of coupled learning (CpL)

$$h_\beta^{\mathrm{CpL}} := \arg\min_h E(\theta, x, h, o_\beta^{\mathrm{CpL}}), \qquad o_\beta^{\mathrm{CpL}} := (1 - \beta)o_\star + \beta y, \tag{52}$$

and the definition of the (positively-perturbed) CpL rule

$$\Delta^{\mathrm{CpL}}\theta := \frac{\eta}{\beta}\left(\frac{\partial E}{\partial \theta}(\theta, x, h_\star, o_\star) - \frac{\partial E}{\partial \theta}\left(\theta, x, h_\beta^{\mathrm{CpL}}, o_\beta^{\mathrm{CpL}}\right)\right). \tag{53}$$

Similar to the contrastive function $\mathcal{L}_{\mathrm{CL}}$ of contrastive learning (10) and the surrogate loss $\mathcal{L}_\beta^{\mathrm{EP}}$ of equilibrium propagation (21), Stern et al. [2022] define the contrast

$$\mathcal{L}_{\mathrm{CpL}}^{(2)} := \frac{1}{\beta}\left(E(\theta, x, h_\beta^{\mathrm{CpL}}, o_\beta^{\mathrm{CpL}}) - E(\theta, x, h_\star, o_\star)\right) \tag{54}$$

and argue that the CpL rule performs gradient descent on $\mathcal{L}_{\mathrm{CpL}}^{(2)}$. Besides, Stern et al. [2021] argue that the CpL rule optimizes both the function

$$\mathcal{L}_{\mathrm{CpL}}^{(1)}(\theta, x, y) := (y - o(\theta, x))^\top \cdot \frac{\partial^2 G}{\partial o^2}(\theta, x, o(\theta, x)) \cdot (y - o(\theta, x)), \quad G(\theta, x, o) := \min_h E(\theta, x, h, o) \tag{55}$$

and the squared error loss

$$\mathcal{L}_{\mathrm{MSE}}(\theta, x, y) := (o(\theta, x) - y)^2, \tag{56}$$

where $o(\theta, x) = o_\star$ is the free output value.

In this appendix, we show that the CpL algorithm does not perform gradient descent on $\mathcal{L}_{\mathrm{CpL}}^{(2)}$ (Appendix B.1). We also show that, in general, the CpL rule optimizes neither the squared error $\mathcal{L}_{\mathrm{MSE}}$, nor the function $\mathcal{L}_{\mathrm{CpL}}^{(1)}$. To this end, we present two examples: in the first one, the function $\mathcal{L}_{\mathrm{MSE}}$ increases after a single step of the coupled learning rule (Appendix B.2) ; in the second one, the function $\mathcal{L}_{\mathrm{CpL}}^{(1)}$ increases (Appendix B.3).

## B.1    Coupled learning does not perform gradient descent on the contrast $\mathcal{L}_{\mathrm{CpL}}^{(2)}$

Stern et al. [2022] note that $\mathcal{L}_{\mathrm{CpL}}^{(2)}$ is non-negative and achieves its minimum possible value ($\mathcal{L}_{\mathrm{CpL}}^{(2)} = 0$) when $o_\star = y$, which motivates them to minimize $\mathcal{L}_{\mathrm{CpL}}^{(2)}$ by gradient descent. However, as we show here, the coupled learning rule does not perform gradient descent on $\mathcal{L}_{\mathrm{CpL}}^{(2)}$. Indeed, while the free state is characterized by

$$\frac{\partial E}{\partial h}(\theta, x, h_\star, o_\star) = 0, \qquad \frac{\partial E}{\partial o}(\theta, x, h_\star, o_\star) = 0, \tag{57}$$

the perturbed state of CpL only satisfies

$$\frac{\partial E}{\partial h}\left(\theta, x, h_\beta^{\mathrm{CpL}}, o_\beta^{\mathrm{CpL}}\right) = 0. \tag{58}$$

The partial derivative $\frac{\partial E}{\partial o}\left(\theta, x, h_\beta^{\mathrm{CpL}}, o_\beta^{\mathrm{CpL}}\right)$ does not vanish in general, in contrast with the perturbed state of equilibrium propagation (EP), which satisfies (31). Thus, we have

$$\frac{d}{d\theta}E(\theta, x, h_\star, o_\star) = \frac{\partial E}{\partial \theta}(\theta, x, h_\star, o_\star) \tag{59}$$

$$+ \underbrace{\frac{\partial E}{\partial h}(\theta, x, h_\star, o_\star)}_{=0} \cdot \frac{\partial h_\star}{\partial \theta} + \underbrace{\frac{\partial E}{\partial o}(\theta, x, h_\star, o_\star)}_{=0} \cdot \frac{\partial o_\star}{\partial \theta} \tag{60}$$

$$= \frac{\partial E}{\partial \theta}(\theta, x, h_\star, o_\star), \tag{61}$$

and

$$\frac{d}{d\theta}E\left(\theta, x, h_\beta^{\mathrm{CpL}}, o_\beta^{\mathrm{CpL}}\right) = \frac{\partial E}{\partial \theta}\left(\theta, x, h_\beta^{\mathrm{CpL}}, o_\beta^{\mathrm{CpL}}\right) \tag{62}$$

$$+ \underbrace{\frac{\partial E}{\partial h}\left(\theta, x, h_\beta^{\mathrm{CpL}}, o_\beta^{\mathrm{CpL}}\right)}_{=0} \cdot \frac{\partial h_\beta^{\mathrm{CpL}}}{\partial \theta} + \underbrace{\frac{\partial E}{\partial o}\left(\theta, x, h_\beta^{\mathrm{CpL}}, o_\beta^{\mathrm{CpL}}\right)}_{\neq 0 \text{ in general}} \cdot \frac{\partial o_\beta^{\mathrm{CpL}}}{\partial \theta}$$

$$\tag{63}$$

$$= \frac{\partial E}{\partial \theta}\left(\theta, x, h_\beta^{\mathrm{CpL}}, o_\beta^{\mathrm{CpL}}\right) + (1-\beta)\underbrace{\frac{\partial E}{\partial o}\left(\theta, x, h_\beta^{\mathrm{CpL}}, o_\beta^{\mathrm{CpL}}\right)}_{\neq 0 \text{ in general}} \cdot \frac{\partial o_\star}{\partial \theta}. $$

$$\tag{64}$$

As a result, the gradient of $\mathcal{L}_{\mathrm{CpL}}^{(2)}$ is

$$\frac{\partial \mathcal{L}_{\mathrm{CpL}}^{(2)}}{\partial \theta} = \frac{1}{\beta}\left(\frac{\partial E}{\partial \theta}(\theta, x, h_\beta^{\mathrm{CpL}}, o_\beta^{\mathrm{CpL}}) - \frac{\partial E}{\partial \theta}(\theta, x, h_\star, o_\star)\right) \tag{65}$$

$$+ \frac{1-\beta}{\beta}\underbrace{\frac{\partial E}{\partial o}\left(\theta, x, h_\beta^{\mathrm{CpL}}, o_\beta^{\mathrm{CpL}}\right) \cdot \frac{\partial o_\star}{\partial \theta}}_{\neq 0 \text{ in general}}. \tag{66}$$

The last term does not vanish in general, unless $\beta = 1$. We note that the case $\beta = 1$ corresponds to contrastive learning (CL, section 2.1) ; hence we have recovered Theorem 1, known since Movellan [1991].

Therefore, contrary to CL that performs gradient descent on $\mathcal{L}_{\mathrm{CL}}$ and EP that performs gradient descent on $\mathcal{L}_\beta^{\mathrm{EP}}$ for any value of $\beta$, the CpL rule does not perform gradient descent on $\mathcal{L}_{\mathrm{CpL}}^{(2)}$ for $\beta \neq 1$.

### B.2 One step of coupled learning may increase the squared error $\mathcal{L}_{\mathrm{MSE}}$

Next, we show in an example that CpL does not optimize $\mathcal{L}_{\mathrm{MSE}}$. Let $A$ be the $2 \times 2$ square matrix

$$A := \begin{pmatrix} 1 & -1 \\ -1 & 2 \end{pmatrix} \tag{67}$$

and $b : \mathbb{R}^2 \to \mathbb{R}^2$ the function defined by

$$b(\theta_1, \theta_2) := \begin{pmatrix} 1 + \theta_2 \\ \theta_1 + 2\theta_2 \end{pmatrix}, \qquad \theta_1, \theta_2 \in \mathbb{R}. \tag{68}$$

We define the energy function $E : \mathbb{R}^2 \times \mathbb{R}^2 \to \mathbb{R}$ by

$$\forall \theta \in \mathbb{R}^2, \forall o \in \mathbb{R}^2, \quad E(\theta, o) := \frac{1}{2}(o - b(\theta))^\top A(o - b(\theta)). \tag{69}$$

For simplicity, there is no input ($x$) and no hidden variable ($h$) in our example. Since the matrix $A$ is symmetric positive definite, the minimum of $o \mapsto E(\theta, o)$ is achieved at the point $o_\star = b(\theta)$, where $E(\theta, o_\star) = 0$. Next, let's assume that the 'desired output' is $y := (0,0) \in \mathbb{R}^2$. The perturbed state of the CpL algorithm is

$$o_\beta^{\mathrm{CpL}} := (1-\beta)o_\star + \beta y = (1-\beta)b(\theta), \tag{70}$$

where $\beta \in \mathbb{R}$ is the nudging parameter, and the P-CpL rule is

$$\Delta^{\mathrm{CpL}}\theta := \frac{\eta}{\beta}\left(\frac{\partial E}{\partial \theta}(\theta, o_\star) - \frac{\partial E}{\partial \theta}(\theta, o_\beta^{\mathrm{CpL}})\right), \tag{71}$$

where $\eta$ is the learning rate. Using (69) we calculate

$$\frac{\partial E}{\partial \theta}(\theta, o) = b'(\theta)^\top A(o - b(\theta)). \tag{72}$$

In particular,

$$\frac{\partial E}{\partial \theta}(\theta, o_\star) = 0 \tag{73}$$

and

$$\frac{\partial E}{\partial \theta}(\theta, o_\beta^{\mathrm{CpL}}) = b'(\theta)^\top A \left(o_\beta^{\mathrm{CpL}} - b(\theta)\right) = -\beta b'(\theta)^\top A b(\theta), \tag{74}$$

where we have used (70). So the P-CpL rule (71) rewrites[8]

$$\Delta^{\mathrm{CpL}}\theta = -\eta b'(\theta)^\top A b(\theta). \tag{75}$$

Next, we define the squared error between the prediction $o_\star$ and the 'desired output' $y = (0, 0)$,

$$\mathcal{L}_{\mathrm{MSE}}(\theta) := \|o_\star - y\|^2 = \|b(\theta)\|^2. \tag{76}$$

To conclude, we show that for $\eta$ small enough, $\mathcal{L}_{\mathrm{MSE}}(\theta_0 + \Delta^{\mathrm{CpL}}\theta) > \mathcal{L}_{\mathrm{MSE}}(\theta_0)$ at the point $\theta_0 = (0, 0)$ independently of the value of the nudging parameter $\beta \in \mathbb{R}$. To show this, first we calculate $\frac{\partial \mathcal{L}_{\mathrm{MSE}}}{\partial \theta} \cdot \Delta^{\mathrm{CpL}}\theta$. We have

$$\frac{\partial \mathcal{L}_{\mathrm{MSE}}}{\partial \theta} = b(\theta)^\top b'(\theta), \tag{77}$$

so that

$$\frac{\partial \mathcal{L}_{\mathrm{MSE}}}{\partial \theta} \cdot \Delta^{\mathrm{CpL}}\theta = -\eta b(\theta)^\top b'(\theta) b'(\theta)^\top A b(\theta). \tag{78}$$

Let us calculate:

$$b'(\theta) = \begin{pmatrix} 0 & 1 \\ 1 & 2 \end{pmatrix}, \tag{79}$$

and

$$b'(\theta) b'(\theta)^\top A = \begin{pmatrix} 0 & 1 \\ 1 & 2 \end{pmatrix} \begin{pmatrix} 0 & 1 \\ 1 & 2 \end{pmatrix} \begin{pmatrix} 1 & -1 \\ -1 & 2 \end{pmatrix} \tag{80}$$

$$= \begin{pmatrix} 1 & 2 \\ 2 & 5 \end{pmatrix} \cdot \begin{pmatrix} 1 & -1 \\ -1 & 2 \end{pmatrix} \tag{81}$$

$$= \begin{pmatrix} -1 & 3 \\ -3 & 8 \end{pmatrix}. \tag{82}$$

Since at the point $\theta_0 = (0, 0)$ we have

$$b(\theta_0) = \begin{pmatrix} 1 \\ 0 \end{pmatrix}, \tag{83}$$

it follows that

$$b(\theta_0)^\top b'(\theta_0) b'(\theta_0)^\top A b(\theta_0) = -1. \tag{84}$$

Therefore, at the point $\theta_0 = (0, 0)$,

$$\frac{\partial \mathcal{L}_{\mathrm{MSE}}}{\partial \theta} \cdot \Delta^{\mathrm{CpL}}\theta = \eta. \tag{85}$$

Finally,

$$\mathcal{L}_{\mathrm{MSE}}(\theta_0 + \Delta^{\mathrm{CpL}}\theta) - \mathcal{L}_{\mathrm{MSE}}(\theta_0) = \frac{\partial \mathcal{L}_{\mathrm{MSE}}}{\partial \theta} \cdot \Delta^{\mathrm{CpL}}\theta + O(\|\Delta^{\mathrm{CpL}}\theta\|^2) = \eta + O(\eta^2). \tag{86}$$

Thus, provided that the learning rate $\eta > 0$ is small enough, the loss value $\mathcal{L}_{\mathrm{MSE}}(\theta_0 + \Delta\theta)$ after one step of coupled learning is larger than $\mathcal{L}_{\mathrm{MSE}}(\theta_0)$. This holds for any nudging value $\beta \in \mathbb{R}$.

---

[8]We note that the learning rules of N-CpL and C-CpL are identical in this example.

## B.3 One step of coupled learning may increase $\mathcal{L}_{\mathrm{CpL}}^{(1)}$

Finally, we show that CpL does not optimize $\mathcal{L}_{\mathrm{CpL}}^{(1)}$. To this end, we consider a second example, where we define the energy function $E : (0, 5/4) \times \mathbb{R} \to \mathbb{R}$ by

$$\forall \theta \in (0, 5/4), \quad \forall o \in \mathbb{R}, \qquad E(\theta, o) := \frac{1}{2}(5 - 4\theta)(o - \theta)^2. \tag{87}$$

The minimum of $o \mapsto E(\theta, o)$ is obtained at the point $o_\star = \theta$, where $E(\theta, o_\star) = 0$. As in the previous example, we assume that the 'desired output' is $y := 0$, so the perturbed state of CpL is

$$o_\beta^{\mathrm{CpL}} = (1 - \beta)o_\star + \beta y = (1 - \beta)\theta \tag{88}$$

and the CpL rule is

$$\Delta^{\mathrm{CpL}}\theta = \frac{\eta}{\beta}\left(\frac{\partial E}{\partial \theta}(\theta, o_\star) - \frac{\partial E}{\partial \theta}(\theta, o_{\mathrm{CL}}^\beta)\right). \tag{89}$$

We calculate

$$\frac{\partial E}{\partial \theta}(\theta, o) = (5 - 2\theta - 2o)(o - \theta). \tag{90}$$

In particular,

$$\frac{\partial E}{\partial \theta}(\theta, o_\star) = 0 \tag{91}$$

and

$$\frac{\partial E}{\partial \theta}(\theta, o_\beta^{\mathrm{CpL}}) = (5 - 2\theta - 2o_\beta^{\mathrm{CpL}})(o_\beta^{\mathrm{CpL}} - \theta) = (5 - 4\theta + 2\beta\theta)\beta\theta, \tag{92}$$

so the CpL rule reads

$$\Delta^{\mathrm{CpL}}\theta = -\eta(5 - 4\theta + 2\beta\theta)\theta. \tag{93}$$

Now, recall the definition of the function $\mathcal{L}_{\mathrm{CpL}}^{(1)}$ of (16). In this example, it is equal to

$$\mathcal{L}_{\mathrm{CpL}}^{(1)}(\theta) := o_\star \cdot \frac{\partial^2 E}{\partial o^2}(\theta, o_\star) \cdot o_\star = (5 - 4\theta)\theta^2, \tag{94}$$

and its derivative (or 'gradient') is

$$(\mathcal{L}_{\mathrm{CpL}}^{(1)})'(\theta) = 10\theta - 12\theta^2. \tag{95}$$

At the point $\theta_0 = 1$ we have $(\mathcal{L}_{\mathrm{CpL}}^{(1)})'(\theta_0) = -2$, and $\Delta^{\mathrm{CpL}}\theta = -\eta(1 + 2\beta)$. This means that $(\mathcal{L}_{\mathrm{CpL}}^{(1)})'(\theta_0) < 0$, and $\Delta^{\mathrm{CpL}}\theta < 0$ for any nudging value $\beta > -1/2$. Thus, for a small $\eta$ and any $\beta > -1/2$, the function $\mathcal{L}_{\mathrm{CpL}}^{(1)}$ increases after a single step of the CpL rule.

## C Energy minimization procedure for deep convolutional Hopfield networks via asynchronous updates

Prior works on deep convolutional Hopfield networks [Ernoult et al., 2019, Laborieux et al., 2021, Laydevant et al., 2021, Laborieux and Zenke, 2022] minimized the energy function (18) using a 'synchronous update' scheme, where at every iteration, all the layers are updated synchronously. In this work, we use an 'asynchronous update' scheme.

Recall the form of the energy function of a deep convolutional Hopfield network (DCHN), given by (18). Ernoult et al. [2019] write the energy function in the form

$$E(\theta, s) = \frac{1}{2} \|s\|^2 - \Phi(\theta, s), \tag{96}$$

where $\|s\|^2 := \sum_{k=1}^{5} \|s_k\|^2$ and $\Phi$ is the so-called 'primitive function', defined as

$$\Phi(\theta, s) := \sum_{k=1}^{4} s_k \bullet \mathcal{P}(w_k \star s_{k-1}) + s_5^\top w_5 s_4 + \sum_{k=1}^{5} b_k^\top s_k. \tag{97}$$

In this expression, $s = (s_0, s_1, s_2, s_3, s_4, s_5)$ is the state of the network, $x = s_0$ is the input layer, $h = (s_1, s_2, s_3, s_4)$ is the hidden variable, $o = s_5$ is the output variable, $w_k$ and $b_k$ are the kernel (the weights) and bias of layer $k$, $\star$ is the convolution operation, $\mathcal{P}$ is the max pooling operation, $\bullet$ is the scalar product for pairs of tensors, and $\theta = \{w_k, b_k \mid 1 \leq k \leq 5\}$ is the set of model parameters. Ernoult et al. [2019] define the dynamics

$$s^{(t+1)} = \sigma\left(\frac{\partial \Phi}{\partial s}(\theta, s^{(t)})\right), \tag{98}$$

where $\sigma(\cdot) = \max(0, \min(\cdot, 1))$ is the 'hard sigmoid function', which they *assume* to converge to a fixed point $s_\star$ characterized by

$$s_\star = \sigma\left(\frac{\partial \Phi}{\partial s}(\theta, s_\star)\right). \tag{99}$$

Importantly, the fixed point $s_\star$ is also a critical point of the energy function, $s \mapsto E(\theta, s)$, since $\frac{\partial E}{\partial s}(\theta, s_\star) = s_\star - \frac{\partial \Phi}{\partial s}(\theta, s_\star) = 0$. This dynamics is what we call here 'synchronous update' scheme, because at every iteration (or 'time step') $t$, all the layers $s_k$ ($1 \leq k \leq 5$) are updated synchronously.

In this work, we use an 'asynchronous update scheme', where at each iteration, we first update the layers of even indices, and then we update the layers of odd indices. Denoting $s_o = (s_1, s_3, s_5)$ the configuration of odd layers, and $s_e = (s_2, s_4)$ the configuration of even layers, the asynchronous update scheme reads

$$s_o^{(t+1)} = \sigma_o\left(\frac{\partial \Phi}{\partial s_o}\left(\theta, s_o^{(t)}, s_e^{(t)}\right)\right), \tag{100}$$

where $\sigma_o$ is the activation function of odd layers, and

$$s_e^{(t+1)} = \sigma_e\left(\frac{\partial \Phi}{\partial s_e}\left(\theta, s_o^{(t+1)}, s_e^{(t)}\right)\right), \tag{101}$$

where $\sigma_e$ is the activation function of even layers. For the layers of index $k$ such that $1 \leq k \leq 4$, the activation function is $\sigma_k(\cdot) = \max(0, \min(\cdot, 1))$, i.e. the 'hard sigmoid function', and $\sigma_5$ is the identity (the 'linear activation function').

In this work, we use the max pooling operation $\mathcal{P}_{\max}$. Although for this choice of pooling operation we do not have a proof of convergence of either of the two schemes ('synchronous' and 'asynchronous'), we find experimentally that the asynchronous scheme converges faster. This allows us to reduce the number of iterations in the free and perturbed phases and thus get a significant speedup, as explained in the next subsection.

In the case where $\mathcal{P}$ is the average (mean) pooling operation, however, we can prove that the energy decreases at each step of the asynchronous update scheme. Indeed, for the average pooling operation $\mathcal{P}_{\text{avg}}$, the primitive $\Phi$ is a linear function of $s_e$, i.e. the energy $E$ is of the form

$$E(s_e) = \frac{1}{2}\|s_e\|^2 - B_e \cdot s_e + C_e \tag{102}$$

for some constants $B_e$ and $C_e$ that do not depend on $s_e$. Specifically

$$B_k := \mathcal{P}\left(w_k \star s_{k-1}^{(t+1)}\right) + w_{k+1}^{\top} \star \mathcal{P}^{-1}\left(s_{k+1}^{(t+1)}\right) + b_k \tag{103}$$

for every layer $k$. It is easily seen that the minimum of such a function $E(s_e)$ in $\mathbb{R}^{\dim(s_2)} \times \mathbb{R}^{\dim(s_4)}$ is obtained at the point $s_e = B_e$. It is also easily seen that the minimum of $E(s_e)$ in $[0,1]^{\dim(s_2)} \times [0,1]^{\dim(s_4)}$ is obtained at the point $s_e = \max(0, \min(B_e, 1))$. Similar expressions hold for the configuration of odd layers $s_o$ that minimizes the energy function given the state of even layers fixed. Given that $\Phi$ is a linear function of $s_o$, we have that $B_o = \frac{\partial \Phi}{\partial s_o}(\theta, s)$, and similarly $B_e = \frac{\partial \Phi}{\partial s_e}(\theta, s)$. Thus, we conclude that the energy function decreases at each step of the asynchronous update scheme (100)-(101).

For the max pooling operation used in this work, the primitive function $\Phi$ is not a linear function of $s_e$ or $s_o$, and the above argument no longer holds.

### C.1   13.5x simulation speedup compared to Laborieux et al. [2021]

We compare in simulations two settings:

1. In the first setting, we use the asynchronous update scheme with 60 iterations at inference, 20 iterations in the perturbed phases, and 16 bit precision. This is the setting of the simulations performed for the comparative study in the present work.

2. In the second setting, we use the synchronous update scheme with 250 iterations at inference, 30 iterations in the perturbed phase, and 32 bits precision. This is (up to minor differences) the setting of the simulations of Laborieux et al. [2021].

We perform the simulations using the C-EP algorithm on CIFAR-10, and we use the hyperparameters reported as 'SOTA results on CIFAR10' in Table 5. We perform three runs in the first setting, and one run in the second setting, and we report the results in Table 3. We see that the first setting is 13.5 times faster than the second setting, requiring only 3 hours and 18 minutes to complete on a single A-100 GPU (compared to the 44 hours 29 minutes required in the second setting). Furthermore, the first setting also performs marginally better in terms of training and test error rates than the second one.

Table 3: Comparison of the performance and the wall-clock-time between the simulation setting of Laborieux et al. [2021] and ours. The simulations of Laborieux et al. [2021] use the synchronous update scheme, in conjunction with 32 bits precision, 250 iterations at inference (free phase), and 30 iterations in the perturbed phase. Our asynchronous update scheme is used in conjunction with 16 bits precision, 60 iterations at inference and 15 iterations in the perturbed phase. Simulations are performed on CIFAR-10 using the C-EP algorithm for training. We perform 100 epochs and use the hyperparameters of the 'SOTA CIFAR-10 simulations' provided in Table 5. We report the test error rate (Test Error), the training error rate (Train Error) and the wall-clock time (WCT). WCT is reported as HH:MM.

| | Test Error | Train Error | WCT |
|---|---|---|---|
| Synchronous - 32 bits - 250 iterations (1 run) | 10.85% | 3.58% | 44:29 |
| Asynchronous - 16 bits - 60 iterations (3 runs) | $10.40 \pm 0.10\%$ | $3.48 \pm 0.14\%$ | 03:18 |

## D Simulation details

**Datasets.** We perform simulations on the MNIST, Fashion-MNIST, SVHN, CIFAR-10 and CIFAR-100 datasets.

The MNIST dataset is composed of images of handwritten digits [LeCun et al., 1998]. Each image $x$ in the dataset is a $28 \times 28$ gray-scaled image and comes with a label $y \in \{0, 1, \ldots, 9\}$ indicating the digit that the image represents. The dataset contains 60,000 training images and 10,000 test images.

The Fashion-MNIST dataset [Xiao et al., 2017] shares the same image size, data format and the same structure of training and testing splits as MNIST. It comprises a training set of 60,000 images and a test set of 10,000 images. Each example is a $28 \times 28$ grayscale image from ten categories of fashion products.

The SVHN dataset [Netzer et al., 2011] contains color images of $32 \times 32$ pixels, derived from house numbers in Google Street View images. Like in MNIST, each image comes with a label corresponding to a digit from 0 to 9. The number of images per class varies due to the natural distribution of digits in the real world. The dataset contains a training set with $73,257$ images, and a test set containing $26,032$ images. The dataset also contains an additional set of $531,131$ other (somewhat less difficult) images, which we do not use in our simulations.

The CIFAR-10 dataset [Krizhevsky et al., 2009] consists of $60,000$ colour images of $32 \times 32$ pixels. These images are split in 10 classes (each corresponding to an object or animal), with $6,000$ images per class. The training set consists of $50,000$ images and the test set of $10,000$ images.

Similarly, the CIFAR-100 dataset [Krizhevsky et al., 2009] also comprises $60,000$ color images with a resolution of $32 \times 32$ pixels, featuring a diverse set of objects and animals. These images are categorized into 100 distinct classes, each containing 600 images. Like CIFAR-10, the dataset is divided into a training set with $50,000$ images and a test set containing the remaining $10,000$ images.

**Data pre-processing and data augmentation.** The data is normalized as in Table 4.

Since the deep convolutional Hopfield network used in our simulations takes as input a 32x32-pixel image, we augment each image of MNIST and Fashion-MNIST to a 32x32-pixel image by adding two pixels at the top, two pixels at the bottom, two pixels to the left and two pixels to the right.

On the training sets of Fashion-MNIST, CIFAR-10 and CIFAR-100, we also use random horizontal flipping. (Random flipping is not used on the training sets of MNIST and SVHN, and it is never used at test time either.)

Table 4: Data normalization. We normalize the input images using the recommended mean ($\mu$) and std ($\sigma$) values for each dataset. The MNIST and Fashion-MNIST images are gray-scale, i.e. they have a unique channel. The SVHN, CIFAR-10 and CIFAR-100 images are color images, i.e. they have three channels.

|  | mean ($\mu$) | std ($\sigma$) |
|---|---|---|
| MNIST | 0.1307 | 0.3081 |
| Fashion-MNIST | 0.2860 | 0.3530 |
| SVHN | (0.4377, 0.4438, 0.4728) | (0.1980, 0.2010, 0.1970) |
| CIFAR-10 | (0.4914, 0.4822, 0.4465) | (3*0.2023, 3*0.1994, 3*0.2010) |
| CIFAR-100 | (0.5071, 0.4867, 0.4408) | (0.2675, 0.2565, 0.2761) |

**Network architecture and hyperparameters.** Table 5 contains the architectural details of the network, as well as the hyperparameters used to obtain the results presented in Table 1, Table 2 and Table 6. The hard-sigmoid activation function is defined as $\sigma(h) := \max(0, \min(h, 1))$.

**Weight initialization.** We initialize the weights of dense interactions and convolutional interactions according to

$$w_{ij} \sim \mathcal{U}(-c, +c), \qquad c = \alpha \sqrt{\frac{1}{\text{fan}_{\text{in}}}}, \tag{104}$$

Table 5: The **top table** indicates the architecture of the deep convolutional Hopfield network (DCHN). We denote $n_\text{in}$ the number of input filters (either 1 or 3 depending on the dataset) and $n_\text{out}$ the number of output units (either 10 or 100 depending on the dataset). The **bottom table** indicates the hyper-parameters used for initializing and training the network. lr means learning rate. The first two columns (Comparative study) provide the values used for the extensive comparison of learning algorithms (Table 1 and Table 6). The other columns (SOTA results) provide the values used to obtain the best performances on MNIST, CIFAR-10 and CIFAR-100, after 100 epochs and 300 epochs (Table 2)

| | Layer shapes | Weight shapes | Bias shapes | Activation functions |
|---|---|---|---|---|
| Layer 0 (inputs) | $n_\text{in}{\times}32{\times}32$ | | | |
| Layer 1 | $128{\times}16{\times}16$ | $128{\times}n_\text{in}{\times}3{\times}3$ | 128 | Hard-sigmoid |
| Layer 2 | $256{\times}8{\times}8$ | $256{\times}128{\times}3{\times}3$ | 256 | Hard-sigmoid |
| Layer 3 | $512{\times}4{\times}4$ | $512{\times}256{\times}3{\times}3$ | 512 | Hard-sigmoid |
| Layer 4 | $512{\times}2{\times}2$ | $512{\times}512{\times}3{\times}3$ | 512 | Hard-sigmoid |
| Layer 5 (outputs) | $n_\text{out}$ | $512{\times}2{\times}2{\times}n_\text{out}$ | 10 | Identity |

| | Comparative study | | SOTA results (Table 2) | | |
|---|---|---|---|---|---|
| | Table 1 | Table 6 | MNIST | CIFAR-10 | CIFAR-100 |
| nudging ($\beta$) | 0.25 | | 0.25 | 0.1 | 0.25 |
| num. iterations at inference ($T$) | 60 | | 60 | 60 | 60 |
| num. iterations at training ($K$) | 15 | | 15 | 20 | 15 |
| gain conv-weight 1 ($\alpha_1$) | 0.5 | 0.7 | 0.5 | 0.4 | 0.5 |
| gain conv-weight 2 ($\alpha_2$) | 0.5 | 0.7 | 0.5 | 0.7 | 0.4 |
| gain conv-weight 3 ($\alpha_3$) | 0.5 | 0.7 | 0.5 | 0.6 | 0.5 |
| gain conv-weight 4 ($\alpha_4$) | 0.5 | 0.7 | 0.5 | 0.3 | 0.8 |
| gain dense-weight 5 ($\alpha_5$) | 0.5 | 0.7 | 0.5 | 0.4 | 0.5 |
| lr conv-weight 1 & bias 1 ($\eta_1$) | 0.0625 | | 0.0625 | 0.03 | 0.03 |
| lr conv-weight 2 & bias 2 ($\eta_2$) | 0.0375 | | 0.0375 | 0.03 | 0.04 |
| lr conv-weight 3 & bias 3 ($\eta_3$) | 0.025 | | 0.025 | 0.03 | 0.04 |
| lr conv-weight 4 & bias 4 ($\eta_4$) | 0.02 | | 0.02 | 0.03 | 0.04 |
| lr dense-weight 5 & bias 5 ($\eta_5$) | 0.0125 | | 0.0125 | 0.03 | 0.025 |
| momentum | 0.9 | | 0.9 | 0.9 | 0.9 |
| weight decay (1e-4) | 3.0 | | 3.0 | 2.5 | 3.5 |
| mini-batch size | 128 | | 128 | 128 | 128 |
| number of epochs | 100 | | 100 | 300    100 | 300    100 |
| $T_\text{max}$ | 100 | | 100 | 300    100 | 300    100 |
| $\eta_\text{min}$ (1e-6) | 2.0 | | 2.0 | 2.0 | 2.0 |

which is the 'Kaiming uniform' scheme rescaled by a factor $\alpha$, that we call the 'gain' here (i.e. a scaling number). For dense weights of shape (size$_\text{pre}$, size$_\text{post}$), we have fan$_\text{in}$ = size$_\text{pre}$ ; for convolutional weights of shape (channel$_\text{in}$, channel$_\text{out}$, height, width), we have fan$_\text{in}$ = channel$_\text{in} \times$ height $\times$ width. See Table 5 for the choice of the gains.

**Energy minimization via asynchronous updates.** To compute the steady state of the network, we use the 'asynchronous update' scheme described in appendix C: at every iteration, we first update the layers of even indices (the first half of the layers) and then we update the layers of odd indices (the other half of the layers). Relaxing all the layers once (first the even layers, then the odd layers) constitutes one 'iteration'. We repeat as many iterations as is necessary until convergence to the steady state. In Table 5, we denote $T$ the number of iterations performed at inference (free phase), and we denote $K$ the number of iterations performed in the perturbed phase.

**Training procedure.** We train our networks with seven energy-based learning (EBL) algorithms, namely: contrastive learning (CL), positively-perturbed equilibrium propagation (P-EP), negatively-perturbed EP (N-EP), centered EP (C-EP), positively-perturbed coupled learning (P-CpL), negatively-perturbed CpL (N-CpL) and centered CpL (C-CpL). Given an EBL algorithm, at each training step of SGD, we proceed as follows. First we pick a mini-batch of samples in the training set, $x$,

and their corresponding labels, $y$. Then we set the nudging parameter to $0$ and we perform a free phase of $T$ asynchronous iterations. This phase allows us in particular to measure the training loss and training error rate for the current mini-batch, to monitor training. We also store the free state $(h_\star, o_\star)$. Next, let's call $\beta_1$ and $\beta_2$ the two nudging values used for training (which depend on the EBL algorithm). First, we set the nudging parameter to $\beta_1$ and we perform a new relaxation phase of $K$ asynchronous iterations. Next, we reset the state of the network to the free state $(h_\star, o_\star)$, then we set the nudging parameter to the second nudging value $\beta_2$, and we perform the last relaxation phase for $K$ asynchronous iterations. Finally, we update all the parameters simultaneously in a single 'parameter update' phase.

**Optimizer and scheduler.** We use mini-batch gradient descent (SGD) with momentum and weight decay. We also use a cosine-annealing scheduler for the learning rates with hyperparameters $T_{\max}$ and $\eta_{\min}$.

**Computational resources.** The code for the simulations uses PyTorch 1.13.1 and TorchVision 0.14.1. Paszke et al. [2017]. The simulations were carried on Nvidia A100 GPUs. For the comparative study, we used five GPUs (one GPU per dataset) and let them run for six days to complete all the simulations.

# E  Additional simulations on SVHN for the comparative study

The results of Table 1 show that most algorithms (P-EP, N-EP, P-CpL and N-CpL) perform very poorly on SVHN: the training process often gets stuck at 81.08% training error and 80.41% test error rate. These results seem to go against our conclusion that N-EP and N-CpL generally perform (much) better than C-EP and C-CpL and CL. We hypothesize that this poor performance on SVHN is due to the network being poorly initialized for this classification task, and we perform additional simulations on SVHN where we change the initialisation scheme of the weights of the network – see Table 5 for the new hyperparameters used for these additional experiments. As for the comparative study of Section 4.1, with this new set of hyperparameters, we compare the seven EBL algorithms of section 2 by training the DCHN of section 3 on SVHN. We also compare the performance of these algorithms to recurrent backpropagation (RBP) and truncated backpropagation (TBP). The results of these additional simulations are reported in Table 6.

Table 6 shows that the N-EP and P-CpL simulations have large variance, so we also report the results for each run. The three runs of N-EP gave the following results: 1) 3.73% test error, 2) 3.87% test error, and 3) 80.41% test error. The third N-EP run was indeed also blocked at 81.08% training error and 80.41% test error (like many other runs). The three runs of P-CpL gave the following results: 1) 71.05% test error, 2) 15.35% test error, and 3) 15.57% test error.

The results of Table 6 are consistent with our conclusions of Section 4.1: 1) strong positive perturbations (CL) yield better results than weak positive perturbations (P-CpL and P-EP), 2) negative perturbations (N-EP, N-CpL) yield better results than positive perturbations (CL, C-CpL, C-EP), 3) two-sided perturbations (C-EP, C-CpL) perform better than one-sided perturbations (N-EP, N-CpL, CL, P-CpL, P-EP), and 4) the EP perturbation technique yields better results than the CpL perturbation technique.

Table 6: Additional experiments on SVHN using a different set of hyperparameters. We compare the seven EBL algorithms and benchmark them against truncated backpropagation (TBP) and recurrent backpropagation (RBP). We report the training and test error rates, averaged over three runs. The hyperparameters used for this study are detailed in Table 5.

|  | SVHN | |
| --- | --- | --- |
|  | Test Error (%) | Train Error (%) |
| TBP | $3.66 \pm 0.05$ | $2.36 \pm 0.04$ |
| RBP | $3.91 \pm 0.05$ | $3.29 \pm 0.04$ |
| CL | $9.64 \pm 1.59$ | $27.08 \pm 5.68$ |
| P-EP | $74.88 \pm 10.60$ | $77.32 \pm 5.84$ |
| N-EP | $29.34 \pm 36.12$ | $29.01 \pm 36.82$ |
| C-EP | $\mathbf{3.58 \pm 0.07}$ | $3.00 \pm 0.05$ |
| P-CpL | $33.99 \pm 26.21$ | $47.28 \pm 26.52$ |
| N-CpL | $4.53 \pm 0.05$ | $5.27 \pm 0.05$ |
| C-CpL | $4.10 \pm 0.04$ | $4.68 \pm 0.10$ |

In Table 7, we report the results of Table 1 together with the test std values.

Table 7: Results of Table 1 reported with the test std values.

| | MNIST | | FashionMNIST | | SVHN | | CIFAR-10 | | CIFAR-100 | |
|---|---|---|---|---|---|---|---|---|---|---|
| | Test | Train | Test | Train | Test | Train | Test | Train | Test | Train |
| TBP | $0.42^{\pm 0.02}$ | 0.23 | $6.12^{\pm 0.10}$ | 3.09 | $3.76^{\pm 0.06}$ | 2.37 | $10.1^{\pm 0.2}$ | 3.1 | $33.4^{\pm 0.5}$ | 17.2 |
| RBP | $0.44^{\pm 0.04}$ | 0.33 | $6.28^{\pm 0.12}$ | 3.70 | $3.87^{\pm 0.09}$ | 3.43 | $10.7^{\pm 0.1}$ | 5.2 | $34.4^{\pm 0.2}$ | 18.2 |
| CL | $0.61^{\pm 0.02}$ | 2.39 | $10.10^{\pm 0.17}$ | 15.49 | $6.10^{\pm 0.13}$ | 15.8 | $31.4^{\pm 2.2}$ | 45.2 | $71.4^{\pm 3.6}$ | 88.6 |
| P-EP | $1.66^{\pm 0.80}$ | 2.29 | $90.00^{\pm 0.00}$ | 89.98 | $83.9^{\pm 4.9}$ | 81.9 | $72.6^{\pm 0.7}$ | 79.5 | $89.4^{\pm 9.5}$ | 95.5 |
| N-EP | $\mathbf{0.42}^{\pm \mathbf{0.01}}$ | 0.19 | $\mathbf{6.22}^{\pm \mathbf{0.10}}$ | 3.87 | $80.4^{\pm 0.0}$ | 81.1 | $11.9^{\pm 0.3}$ | 6.2 | $44.7^{\pm 0.3}$ | 40.1 |
| C-EP | $0.44^{\pm 0.03}$ | 0.20 | $6.47^{\pm 0.13}$ | 4.01 | $\mathbf{3.51}^{\pm \mathbf{0.07}}$ | 3.01 | $\mathbf{11.1}^{\pm \mathbf{0.1}}$ | 5.6 | $\mathbf{37.0}^{\pm \mathbf{0.1}}$ | 26.0 |
| P-CpL | $0.66^{\pm 0.13}$ | 0.59 | $64.70^{\pm 35.77}$ | 65.31 | $40.1^{\pm 28.6}$ | 50.8 | $46.9^{\pm 3.8}$ | 57.7 | $77.9^{\pm 0.5}$ | 91.0 |
| N-CpL | $0.50^{\pm 0.04}$ | 0.88 | $6.86^{\pm 0.05}$ | 6.27 | $80.4^{\pm 0.0}$ | 81.1 | $13.5^{\pm 0.3}$ | 10.2 | $51.9^{\pm 1.3}$ | 50.6 |
| C-CpL | $0.44^{\pm 0.01}$ | 0.38 | $6.91^{\pm 0.11}$ | 5.29 | $4.23^{\pm 0.09}$ | 5.05 | $14.9^{\pm 0.1}$ | 14.6 | $46.5^{\pm 0.1}$ | 37.9 |

