# OpenReview forum: "Energy-based learning algorithms for analog computing: a comparative study"
_NeurIPS.cc/2023/Conference — NeurIPS 2023 poster_

### Official Review · Reviewer_AFqD · 2023-07-05

**Soundness:** 3 good
**Presentation:** 3 good
**Contribution:** 2 fair
**Rating:** 5
**Confidence:** 3

**Summary:**

This paper aims to investigate existing Energy-based learning algorithms on equal footing with same models and datasets. Energy-based algorithms include contrastive learning (CL), equilibrium propagation (EP) and coupled learning (CpL) have been carried out for comparison. The experiments conducted based on deep Hopfield networks (DHNs) show that the centered variant of EP is the best-performing algorithm.

**Strengths:**

This work provides systematic comparison of the different energy-based learning algorithms. The idea is meaningful and the presentation is satisfying.

**Weaknesses:**

it does not present new solutions for efficient energy design/ learning, the generalizability of the comparison (not limited on DCHN) should be further discussed.

**Questions:**

1.	In this paper, what’s the reason that deep convolutional Hopfield network (DCHN) is suitable for generic comparison? The authors should present detailed illustrations.
2.	The experimental results show that equilibrium propagation performs better than coupled learning. Does this conclusion hold true for large-scale data training? The authors need to present deeper analyses or proofs.
3.	It seems that this paper does not provide new solutions for efficient energy design/ learning. According to the conclusion presented in this paper, how can we design powerful EBL models with more complex network architectures?

**Limitations:**

See weaknesses.

---

> ### Author Rebuttal · Authors · 2023-08-09
>
> We have explained in the common rebuttal the novelties in our work. In particular, our work introduces a novel asynchronous update scheme for accelerating the convergence of the energy-minimization process in DCHNs. This asynchronous update scheme allows us to achieve a 13.5x speedup compared to Laborieux et al. (2021) in our overall simulations. (We refer to Appendix C in the supplementary material for details.)
>
> We have also clarified in the common rebuttal why we chose DCHNs for comparing the different training algorithms. Among the energy-based architectures compatible with analog computing, we note that DCHNs achieve the highest reported accuracies for energy-based learning algorithms on image-recognition tasks, which is the focus of this work. While other architectures have not been studied, we believe that DCHNs are the best choice for this reason.
>
> We agree that there is no obvious reason why DCHNs should be generic for the comparison of EBL algorithms, and that we have no evidence that our findings would generalize beyond this setting. We will explain in the conclusion section that our experimental study is limited to DCHNs and that our findings might not hold in the context of other network models. However, we note that our Theorems 2 and 3 indicate that our findings are likely to generalize beyond the setting of DCHNs.
>
> Finally, while there is no certainty that our findings would hold on large-scale data, we emphasize that our comparative study is the very first such study for comparing EBL algorithms.

---

> > ### Author Response · Authors · 2023-08-17
> > **Engaging in a discussion with Reviewer AFqD**
> >
> > Dear Reviewer AFqD,
> >
> > Thank you for your time in reviewing our work. As the discussion period is on-going, we would be happy to address any remaining question you may have in the light of our rebuttal.
> >
> > We understand from your review that you raised concerns about:
> >
> > (1) the novelty of our work
> >
> > (2) the choice of DCHNs for comparing the seven learning algorithms
> >
> > (3) the generalization of our empirical findings to other models and large scale data
> >
> > We believe that we have addressed these three points in our response from August 9. For your reference, here is a summary of our response from August 9 that addresses these three points:
> >
> > (1) Our work introduces a novel energy minimization algorithm (the “asynchronous update procedure”, which yields a 13.5x speedup compared to Laborieux et al, 2021), and three novel energy-based learning algorithms, namely NEP, NCpL and CCpL. Our study revealed that our NEP algorithm performs much better than the original PEP algorithm introduced by Scellier and Bengio (2017), and often performs as well as the CEP algorithm introduced by Laborieux et al (2021). Similarly, our study revealed that both our CCpL and NCpL algorithms perform better than the original PCpL algorithm introduced by Stern et al (2021). Kindly note that after reading our rebuttal, Reviewer dBi8, who shared similar concerns as yours regarding the novelty of our work, increased his/her score by recommending us to "emphasiz[e] that NEP, NCpL and CCpL are novel contributions of this work more prominently in revisions" and "agree[s] that these are useful and novel variants which yield counter-intuitive insights into EBL learning".
> >
> > (2) Our work is focused on models relying on local update rules, both for the layers’ activation and for the weights. Our motivation is that such models are promising for the development of low-power hardware for AI (analog chips, as opposed to digital chips such as GPUs). Our choice to use DCHNs for comparing the seven energy-based learning algorithms is that, among the energy-based models relying on local update rules, DCHNs are to date the best performing architectures.
> >
> > (3) While we have no proof that our empirical findings will generalize beyond DCHNs and to larger datasets, our novel Theorems 2 and 3 tell us that this is likely to be the case.
> >
> > We thank you again for your time and remain at your disposal for any further questions you may have.

---

> > ### Comment · Reviewer_AFqD · 2023-08-18
> > **Rebuttal Response**
> >
> > Thanks for the author’s response to my questions. In the rebuttal stage, the authors have addressed most of my concerns and clarify the contribution of this work. This paper is well-written and organized, but the contribution is not very significant, so I will keep my rating score.

---

> > > ### Author Response · Authors · 2023-08-21
> > >
> > > Dear Reviewer AFqD,
> > >
> > > Thank you for your positive comment about the quality of the writing and of the organization of the paper.

---

### Official Review · Reviewer_5nrK · 2023-07-05

**Soundness:** 3 good
**Presentation:** 2 fair
**Contribution:** 2 fair
**Rating:** 5
**Confidence:** 2

**Summary:**

The paper focuses on exploring and comparing various energy-based training methods for deep convolutional Hopfield networks. The performance of contrastive learning, positively-perturbed, negatively-perturbed, and centered versions of Equilibrium propagation and Coupled learning algorithms are evaluated. The paper establishes state-of-the-art (SOTA) results for some of these algorithms.

**Strengths:**

The set of experiments conducted in this paper provides a comprehensive comparison of the performance of different training methods on a variety of datasets. This extensive evaluation contributes to a robust understanding of the performance of deep convolutional Hopfield networks (DCHN) and their applicability across different datasets.

**Weaknesses:**

1) One issue with this paper is the lack of citation and discussion of recent works on Energy Based Models (EBMs) that are relevant to the topic, such as Grathwohl et al. (2019), Nijkamp et al. (2019b;a), and Du & Mordatch (2019). These papers analyze the application of EBMs to image classification tasks, propose different training techniques, and compare their performance. It would be beneficial to include a discussion of these works and how they relate to the research presented in this paper.

2) Subsection 2.5, which presents the theoretical results, appears disconnected from the main body of the paper. There is no discussion of the implications of these theorems or how they relate to the research findings. It is important to establish a clear connection between this theoretical section and the rest of the paper to provide a cohesive narrative.

**Questions:**

What factors motivated the authors to choose the Deep Convolutional Hopfield Network (DCHN) model as the framework for comparing the training methods in the paper?

**Limitations:**

The paper does not include a discussion on the limitations.

---

> ### Author Rebuttal · Authors · 2023-08-09
>
> First, we thank the reviewer for their remark on recent works on energy-based models (EBMs). We understand that the term “energy-based model” is ambiguous and refers to different lines of works with very different motivations, which we would like to clarify here. For clarity, we start by highlighting the common points between our work and the referenced works, and then we discuss the differences and implications of these differences.
>
> Both in our work and in the referenced works, the model is defined by an energy function, which is a scalar function of the weights and activations of the network. Inference and training in these EBMs involve minimizing this energy function in the activation space. For instance, in Grathwohl et al (2020), they perform gradient descent in the input space:
> $$ x \leftarrow x - \alpha \partial_x E(f_\theta(x)) $$
> where $f_θ(x)$ represents the model logits given an input $x$. Similarly, in our work we minimize an energy function (the Hopfield energy) thanks to our “asynchronous update procedure” in the activation space.
>
> There is, however, a crucial difference between our work and the referenced works. Our work is concerned with models in which inference and learning are achieved using local update rules, with the long-term motivation of building new, highly-efficient (low-power), hardware for AI. The whole purpose of our line of work is to totally obviate the use of the backpropagation algorithm and perform model optimization (inference and learning) by leveraging locally computed quantities, with the longer term goal in mind to design energy-efficient processors dedicated to model optimization.
> In our work, the asynchronous update procedure to minimize the energy function of a Hopfield network reads:
>
> $$ s_i \leftarrow \sum_j w_{ij} s_j + b_i $$
>
> and the learning rule of CL/EP/CpL in such Hopfield networks reads:
>
> $$\Delta w_{ij} \propto \left( s_i^{\rm perturbed} s_j^{\rm perturbed} -  s_i^{\rm free} s_j^{\rm free} \right)$$
>
> Importantly, both the energy minimization procedure and the learning rule require solely locally available information to update the state variables $s_i$ and weights $w_{ij}$. This feature makes our model amenable to highly efficient implementation on dedicated (analog) hardware. Yi et al (2023) have shown that such Hopfield networks can be built in memristive networks (analog networks) and trained using 10,000x less energy compared to DNNs trained on GPUs.
>
> In the referenced works, the motivation is very different from ours. For example, Grathwohl et al (2020) aim to scale EBM training to build large, well-calibrated and adversarially robust discriminative and generative models. Their algorithms do not preclude the use of the backpropagation algorithm, which they use not only for parameter gradient computation, but also to run stochastic gradient Langevin dynamics. Because they make use of the backpropagation algorithm, which in turn requires that their models runs on digital processors such as GPUs, it is unclear if their model could be useful to build low-power hardware for AI, which is the motivation of our work.
>
> To avoid the confusion between the EBMs of the referenced works and the models considered in our work, we propose to change the name of our approach to “energy-driven models” instead of “energy-based models”, and to change the title to “Energy-driven learning algorithms for analog computing: a comparative study”. We will also explain in the introduction section that the energy-driven approach of our work differs from this other line of work on energy-based models. We hope that this clarifies the motivation of our work.
>
>
> Second, we respectfully disagree with the reviewer’s remark that “Section 2.5, which presents the theoretical results, appears disconnected from the main body of the paper”. We discuss the implications of our theorems and how they relate to the research findings in the discussion section (section 4.2 of the paper). For example, we write: “algorithms employing a positive perturbation (P-EP and P-CpL) perform significantly worse than those employing a negative perturbation (N-EP and N-CpL). [...] Theorem 2 sheds light on this observation: N-EP optimizes an upper bound of the cost function, whereas P-EP optimizes a lower bound”. As pointed out by reviewer dBi8, “Theorems 2 and 3 very nicely corroborate the empirical observations”.
>
>
> Finally, our motivation for choosing DCHNs for comparing the different learning algorithms is that DCHNs are suitable for building low-power AI. As explained in the common rebuttal (see above), among all energy-based architectures compatible with analog hardware, DCHNs are the best-performing models to date.
>
> Reference:
> Yi, S. I., Kendall, J. D., Williams, R. S., & Kumar, S. (2023). Activity-difference training of deep neural networks using memristor crossbars. Nature Electronics, 6(1), 45-51.

---

> > ### Comment · Reviewer_5nrK · 2023-08-17
> >
> > I appreciate the detailed clarification provided by the authors. It has come to my understanding that a misconception on my part regarding the terminology "energy-based models" has contributed to the confusion. I strongly recommend that the changes to the paper's title and the introduction, as outlined in the rebuttal, be incorporated into the final version of the manuscript. I have opted to enhance my rating by 2 points, resulting in a revised score of 5.

---

> > > ### Author Response · Authors · 2023-08-21
> > >
> > > Dear Reviewer 5nrK,
> > >
> > > Many thanks for acknowledging the differences between the scope of our work (energy-based algorithms relying on local update rules) and the literature on EBMs that relies on backpropagation, and for increasing your score. In case of acceptance, we will amend the title and the introduction to emphasize these differences, as suggested.

---

> ### Author Response · Authors · 2023-08-16
> **We would like to engage in a constructive discussion about our submission**
>
> Dear Reviewer 5nrK,
>
> Thank you for your time. As the authors/reviewers discussion period is on-going, we would be very grateful if you gave us the opportunity to engage in a constructive discussion about our submission. From your review, we understand that you raised concerns about:
>
> (1) our theoretical section
>
> (2) the relation of our work to other works on energy-based models
>
> (3) the choice of DCHNs for comparing the seven learning algorithms
>
> We believe that we addressed your concerns in our response from August 9. Here is a summary of our response:
>
> (1) Our theoretical results (Theorems 2 and 3) corroborate our empirical findings: NEP outperforms PEP, and CEP is the best performing algorithm. Reviewer dBi8 also noted that our theoretical results nicely corroborate our empirical results.
>
> (2) The motivation of our work is that we are interested in local update rules (both for the layers’ activations and for the weights) for analog computing. This motivation is explained in the abstract and the introduction section of our manuscript, and also noted by Reviewer JWNX: “Investigating energy-based learning algorithms is interesting, especially for their compatibility with analog (post-digital) hardware.” We have clarified that models such as the one of Grathwohl et al (2020) differ from the motivation of our work because they rely on global differentiation (backpropagation) rather than locally computed quantities, and therefore are not directly connected to our line of works.
>
> (3) Our choice to use DCHNs for comparing the seven learning algorithms is that, among the models relying on local update rules, DCHNs are to date the best performing architectures.
>
> From these clarifications, we would like to ask if you still have other doubts based on which you recommend rejection?
>
> We thank you again for your time.

---

### Official Review · Reviewer_JWNX · 2023-07-06

**Soundness:** 4 excellent
**Presentation:** 4 excellent
**Contribution:** 3 good
**Rating:** 6
**Confidence:** 4

**Summary:**

This work conducts an extensive comparison of several energy-based learning (EBL) algorithms, including contrastive learning (CL), equilibrium propagation (EP) and coupled learning (CpL). Depending on the type of perturbation used, 9 variants of EP and CpL are examined. Deep Hopfield networks (DHNs) on five vision tasks (MNIST, F-MNIST, SVHN, CIFAR-10 and CIFAR-100) are trained, and different EBL algorithms are compared.

**Strengths:**

Investigating energy-based learning algorithms is interesting, especially for their compatibility with analog (post-digital) hardware. The paper is generally well-written. The concepts and algorithms are mostly clearly presented. Experiments are detailed with good analysis.

**Weaknesses:**

Not clear that Theorem 1-3 are new, or known from existing literatures.

The paper is good for a comparison study of different EBL algorithms, but the algorithm contribution is limited.

The performance of EBL-trained DCHNs on vision tasks (e.g., CIFAR-10) are far behind modern neural networks. It would be better to add such information and comparison.
https://paperswithcode.com/sota/image-classification-on-cifar-10

**Questions:**

see above.

**Limitations:**

see above.

---

> ### Author Rebuttal · Authors · 2023-08-09
>
> Regarding the question on the novelty of the theorems, we would like to clarify that Theorem 1 isn’t new ; it is presented and proved e.g. in Movellan (1991).
> On the other hand, Theorems 2 and 3 are new in the literature: no prior work had shown that EP (resp. CEP) performs gradient descent on the surrogate loss function L_beta (resp. L_{-beta,+beta}).
>
> We have explained in the common rebuttal the algorithmic novelties in our work. While the network architecture (the deep convolutional Hopfield network, DCHN) isn’t novel, we introduce in our work three novel EBL algorithms (NEP, NCpL and CCpL, to compute the parameter gradients) and a novel energy minimization algorithm (the “asynchronous update method” to compute the equilibrium states of DCHNs). Importantly, our novel energy minimization algorithm leads to a 13.5x speedup compared to Laborieux et al. (2021).
>
> Finally, we agree that the performance of DCHNs on the tasks considered in this work lags behind that of SOTA deep learning models. We propose to report the results of SOTA algorithms for comparison, as suggested by the reviewer.

---

> > ### Author Response · Authors · 2023-08-17
> > **Engaging in a discussion with Reviewer JWNX**
> >
> > Dear Reviewer JWNX,
> >
> > Thank you again for the time already spent on reviewing our work. As the discussion period is on-going, we would be very grateful to be given the opportunity to address any remaining concern you may have after reading our rebuttal. Kindly note that after reading our rebuttal, Reviewer dBi8, who shared similar concerns as yours, decided to raise his/her score by recommending us to "emphasiz[e] that NEP, NCpL and CCpL are novel contributions of this work more prominently in revisions" and "agree[s] that these are useful and novel variants which yield counter-intuitive insights into EBL learning". Let us know if we can further complete our answer.

---

> > > ### Comment · Reviewer_JWNX · 2023-08-18
> > > **After reading feedback**
> > >
> > > Thanks for the feedback from the authors.
> > >
> > > The explanation provided in the rebuttal about the novelty  of Theorem 1-3, and the results of SOTA algorithms for comparison, should be added in the main text.
> > >
> > > I can see the algorithmic novelties in this work, which, however, may not be so big, especially considering the lags behind that of SOTA models. So I tend to keep my score.

---

> > > > ### Author Response · Authors · 2023-08-21
> > > >
> > > > Dear Reviewer JWNX,
> > > >
> > > > Thank you for your comments. We would like to add that our work is SOTA within the scope of the literature of hardware-friendly training of DHNs (using local update rules for the units and weights). Matching SOTA vision models in the deep learning literature (e.g. ResNets, VITs) will require to scale up our approach to deeper architectures, which is indeed a very important direction of research.

---

### Official Review · Reviewer_dBi8 · 2023-07-07

**Soundness:** 3 good
**Presentation:** 3 good
**Contribution:** 2 fair
**Rating:** 6
**Confidence:** 3

**Summary:**

This work reviews and compares recent EBL methods on classic image classification benchmarks. The paper first reviews a variety of EBL methods including CL, EP, P-EP, N-EP, C-EP, CpL, P-CpL, N-CpL, and C-CpL. Next, two theorems are presented which show that P/N/C EP approximate gradient descent on the cost function. The experiment section presents a large-scale comparison of the EBL methods along with Backprop methods for image classification on image datasets whose complexity ranges from MNIST to CIFAR-100. The study finds that N-type EBL variants outperform P-type EBL variants, and that C-EP performs best overall. The study also finds that MNIST is not a suitable dataset for identifying differences between method outcomes, and that more complex datasets like CIFAR-100 reveal important differences. A discussion of possible explanations for the results is presented.

**Strengths:**

* The paper is a very useful resource for understanding recent EBL methods.
* Theorems 2 and 3 very nicely corroborate the empirical observations about the effectiveness of P-type over N-type, and the overall superiority of C-type.
* A large-scale study of EBL methods gives crucial context for understanding which EBL techniques are useful and which are not. The findings could serve to motivation directions for developing improved EBL algorithms.

**Weaknesses:**

* The primary weakness of this paper is a lack of novelty. It provides a very useful survey of the literature and a much-needed large scale comparison of EBL methods. However, no new techniques are proposed, and the experimental evaluation is somewhat routine.

**Questions:**

* Can the ideas in this paper be used to propose a new EBL algorithm which draws from the useful properties of the best current models?

**Limitations:**

Limitations are not discussed,

---

> ### Author Rebuttal · Authors · 2023-08-09
>
> We have addressed the novelty factor in the common rebuttal (see above). We would like to emphasize that our asynchronous update method to compute equilibrium states of DCHNs (i.e. to compute minima of the energy function) is novel, and that this technique unlocked significant speedup – indeed, a 13.5x speedup compared to Laborieux et al (2021). We refer to Appendix C in the supplementary material for details.
>
> We have also clarified in the common rebuttal that NEP, NCpL and CCpL are novel, and that  these EBL algorithms are tested for the first time in our work. We hope that this answers the reviewer’s question about new EBL algorithms.
>
> We have also explained in the common rebuttal that we will discuss the limitations of our comparative study (limited to DCHNs) in the conclusion section of our manuscript.
>
> Finally, since the reviewer appreciates the usefulness of our large-scale study of EBL methods, we would like to point out that the significant (13.5x) speedup enabled by our novel “asynchronous update method” is also the reason why our extensive comparative study was possible at all. We carried out 135 simulations (5 datasets * 9 algorithms * 3 runs) on five A100 GPUs for about one week, each run (100 epochs) taking between 3 and 5 hours. While not impossible in theory, without this speedup the same study conducted on 5 A100 GPUs would have taken 3 months to complete (instead of one week). Thus, our novel asynchronous update method (one of the essential novelties of our work) played an important role in making this study possible/affordable.

---

> > ### Comment · Reviewer_dBi8 · 2023-08-17
> > **Thanks for the responses. I will raise my score.**
> >
> > Thanks to the authors for their detailed and informative response. I would recommend emphasizing that NEP, NCpL and CCpL are novel contributions of this work more prominently in revisions, but I agree that these are useful and novel variants which yield counter-intuitive insights into EBL learning. I also appreciate that implementation efficiency is a key part of viable machine learning methods and the author's claims about the technical importance of their work seem reasonable. Overall, the responses motivated me to raise my score to 6.

---

> > > ### Author Response · Authors · 2023-08-21
> > >
> > > Dear Reviewer dBi8,
> > >
> > > Thank you for considering our arguments about the novelty of our work and for raising your score.

---

### Author Rebuttal · Authors · 2023-08-09

We thank the reviewers for their time and comments. Several criticisms of the manuscript by the reviewers fall into the following two categories:

1. Lack of novelty

2. The choice of DCHNs for comparing the seven EBL algorithms, and the lack of discussion of the limitations of our study

We address 1 and 2 in this order.


To address 1, we would like to point out the following novelties in our work.

First, while the network architecture that we consider (the deep convolutional Hopfield network, DCHN) is not novel, our energy minimization method for DCHN (the “asynchronous update method”) is a novel contribution to the literature, as opposed to the “synchronous update method” used e.g. in Laborieux et al (2021). Thanks to our asynchronous update method, using reduced precision (16 bits instead of 32 bits) and fewer iterations (60 iterations instead of 250), we get an overall 13.5x speedup compared to Laborieux et al (2021). We refer to Appendix C in the supplementary material for details. Importantly, the code of Laborieux et al (2021) yields the expected performance only with 32 bit precision ; when running their code with 16 bit precision the performance collapses! This observation suggests that switching from their “synchronous update method” to our “asynchronous update method” improves the quality of convergence to equilibrium (the minimum of the energy function), which in turn allows us the use of 16 bits and reduced number of iterations. Thus, our “asynchronous update method” was critical to get the overall 13.5x speedup.

Second, our work introduces novel training algorithms (EBL algorithms), namely NEP, NCpL and CCpL. NEP is new and tested in our work for the first time. Our study revealed the not obvious fact that our NEP algorithm performs much better than the original PEP algorithm introduced by Scellier and Bengio (2017), and often performs as well as the CEP algorithm introduced by Laborieux et al (2021). Similarly, while Coupled Learning was introduced by Stern et al (2021) in the positively-perturbed version (PCpL), our work introduces CCpL and NCpL and tests them for the first time. Our study reveals that both CCpL and NCpL perform better than the original PCpL algorithm.

Additionally, we provide new theoretical results (Theorems 2 and 3) on how these different EBL algorithms optimize objective functions. In particular, we show that N-EP optimizes an upper bound on the cost function, whereas P-EP optimizes a lower bound.


Next, we address 2. The reason we chose DCHNs for comparison in this manuscript is that, among the energy-based architectures compatible with analog hardware (memristive networks), DCHNs are the best performing architectures in the literature. Given this, and the fact that these are some of the largest scale simulations of these learning algorithms to date, we believe that these results will scale to larger datasets and more complex architectures. We will explain in the conclusion section that our experimental findings are limited to DCHNs, that they might not hold beyond these networks, and that this will be left for future work to investigate.


References:

1. Laborieux, A., Ernoult, M., Scellier, B., Bengio, Y., Grollier, J., & Querlioz, D. (2021). Scaling equilibrium propagation to deep convnets by drastically reducing its gradient estimator bias. Frontiers in neuroscience, 15, 633674.

2. Scellier, B., & Bengio, Y. (2017). Equilibrium propagation: Bridging the gap between energy-based models and backpropagation. Frontiers in computational neuroscience, 11, 24.

3. Stern, M., Hexner, D., Rocks, J. W., & Liu, A. J. (2021). Supervised learning in physical networks: From machine learning to learning machines. Physical Review X, 11(2), 021045.

---

### Decision · Program_Chairs · 2023-09-21

**Decision:**

Accept (poster)

**Comment:**

The paper received mixed ratings before the rebuttal. The reviewers raised shared concerns regarding the clarity of the contribution and the model performance. The authors’ response clarified some of the shared concerns and also steered the reviewer with the most negative opinion about the paper towards positive.

The ACs agree with the reviewers and appreciate the paper’s effort in providing a systematic and clearly presented study of energy-based learning algorithms, thus recommending acceptance. As suggested by reviewers and promised in the authors’ rebuttal, in the final version, the clarity should be further improved by incorporating well-articulated arguments to emphasize the contribution and difference with EBMs.